# The evolutionary history of human spindle genes includes back-and-forth gene flow with Neandertals

**Stéphane Peyrégne\*, Janet Kelso, Benjamin M Peter, Svante Pääbo\***

Department of Evolutionary Genetics, Max Planck Institute for Evolutionary Anthropology, Leipzig, Germany

**Abstract** Proteins associated with the spindle apparatus, a cytoskeletal structure that ensures the proper segregation of chromosomes during cell division, experienced an unusual number of amino acid substitutions in modern humans after the split from the ancestors of Neandertals and Denisovans. Here, we analyze the history of these substitutions and show that some of the genes in which they occur may have been targets of positive selection. We also find that the two changes in the kinetochore scaffold 1 (KNL1) protein, previously believed to be specific to modern humans, were present in some Neandertals. We show that the *KNL1* gene of these Neandertals shared a common ancestor with present-day Africans about 200,000 years ago due to gene flow from the ancestors (or relatives) of modern humans into Neandertals. Subsequently, some non-Africans inherited this modern human-like gene variant from Neandertals, but none inherited the ancestral gene variants. These results add to the growing evidence of early contacts between modern humans and archaic groups in Eurasia and illustrate the intricate relationships among these groups.

**\*For correspondence:**
stephane_peyregne@eva.mpg.de (SPe);
paabo@eva.mpg.de (SPä)

**Competing interest:** The authors declare that no competing interests exist.

## Editor's evaluation

Peyrégne et al., studied the genes encoding proteins of the spindle apparatus which exhibit an elevated number of nonsynonymous substitutions in modern humans. In one of these genes (*KNL1*), comparisons of modern and archaic humans identify that some Neanderthals carried the modern human haplotype at the *KNL1* gene, raising the possibility that Neanderthals acquired it from modern humans. Based on the patterns observed in this gene and estimates of the time to the most recent common ancestor, the authors propose an evolutionary scenario that includes an introgression event from modern humans into Neanderthals around 200,000 years ago, and a more recent introgression event from Neanderthals into non-African populations. This study highlights how inspecting individual genomic regions can reveal a complex history of interactions between modern and archaic humans.

## Introduction

The ancestors of Neandertals and Denisovans diverged from those of modern humans between 522 and 634 thousand years ago (kya) (*Prüfer et al., 2017*). Differences between modern humans and Neandertals, Denisovans and other hominins can therefore reveal traits that changed in modern humans over the past half-million years and are specific to modern humans. The paleontological and archaeological records provide information about morphological and cultural traits (*Trinkaus, 2006*; *Roebroeks and Soressi, 2016*), yet other traits remain inaccessible by such approaches. The retrieval of DNA from archaic human remains and the reconstruction of Neandertal and Denisovan genomes

(*Green et al., 2010*; *Meyer et al., 2012*; *Prüfer et al., 2014*) offer an additional approach to understanding what sets modern and archaic humans apart.

Because large numbers of present-day human genomes have been sequenced while few archaic genomes are available, the comparisons of Neandertal and Denisovan genomes are particularly useful for identifying those nucleotide changes that occurred on the modern human lineage and reached fixation (or high frequencies) in present-day people. According to one estimate (*Prüfer et al., 2014*), there are 31,389 such differences of which only 96 change the amino acid sequence of 87 proteins. Three of these missense changes occur in a single gene, *sperm associated antigen 5* (*SPAG5*), which encodes a protein associated with the spindle apparatus (*Mack and Compton, 2001*; *Chang et al., 2001*). The spindle is a cytoskeletal structure composed of microtubules and associated proteins that attach to the chromosomes and ensures their proper segregation during cell division (*Prosser and Pelletier, 2017*). *SPAG5* is the only gene in the genome with three missense changes, but four of the 87 genes carry two missense changes. One of these is *kinetochore scaffold 1* (*KNL1*, previously *CASC5*) which encodes a protein in the kinetochore (*Cheeseman et al., 2008*), a protein structure at the centromere of chromosomes to which the spindle attaches during mitosis and meiosis. The occurrence of three missense changes in *SPAG5* and two in *KNL1*, as well as one in the gene *KIF18A*, which encodes a protein involved in the movement of chromosomes along microtubules (*Mayr et al., 2007*), is intriguing as it suggests that components of the spindle may have been subject to natural selection during recent human evolution (*Pääbo, 2014*; *Kuhlwilm and Boeckx, 2019*).

We therefore set out to study genes with missense changes that are associated with the spindle apparatus and to reconstruct their evolutionary history in modern and archaic humans. We show that multiple spindle genes may have experienced positive selection on the modern human lineage. We also show that the variant of *KNL1* that carries two derived missense mutations was transferred by gene flow from modern humans to early Neandertals and then again from late Neandertals to modern humans.

## Results
### Spindle-associated genes with missense changes in modern humans

According to the classification of genes in the Gene Ontology database (*Ashburner et al., 2000*; *Gene Ontology Consortium, 2021*) that represents current knowledge about the function of genes, there are eight spindle-associated genes among the 87 genes that carry missense changes that are fixed or almost fixed among present-day humans. These eight genes are *SPAG5* and *KNL1*, with three and two changes, respectively, and *ATRX*, *KATNA1*, *KIF18A*, *NEK6*, *RSPH1*, and *STARD9*, which each carry one missense change. A total of 11 missense changes accumulated in these eight genes. Given the length of spindle-associated genes, six missense changes are expected making this a significant enrichment of amino acid changes compared to random expectation (one-tailed permutation test p=0.045; see Methods). Multiple missense changes in a single gene are also more than expected by chance (one-tailed p=0.044 for two changes; one-tailed p<0.001 for three changes). Note that it is the high number of changes in these genes that stands out and not the number of genes, because eight spindle-associated genes carrying at least one missense change is not more than expected (one-tailed p=0.282). The genes with single changes have a range of functions associated with cell division. ATRX is a chromatin remodeler required for normal chromosome alignment, cohesion and segregation during meiosis and mitosis (*Ritchie et al., 2014*; *Ramamoorthy and Smith, 2015*). KATNA1 and KIF18A are depolymerases that disassemble microtubules in the spindle apparatus and are essential for proper chromosome alignment at the equator of dividing cells (*Mayr et al., 2007*; *Roll-Mecak and McNally, 2010*). NEK6 is a kinase that is required for cell cycle progression through mitosis (*Yin et al., 2003*) and interacts with proteins of the centrosome (*Vaz Meirelles et al., 2010*), an organelle that serves as a microtubule-organizing center and is crucial for the spindle apparatus. RSPH1 localizes to the spindle and chromosomes during meiotic metaphase (*Tsuchida et al., 1998*), when the chromosomes align in the dividing cell. Finally, STARD9 localizes to the centrosome during cell division and is required for spindle assembly by maintaining centrosome integrity (*Torres et al., 2011*). We describe the location and predicted effects of the missense changes in *Appendix 1—tables 1 and 2*.

## Evolutionary history of spindle proteins in modern humans

The enrichment of missense changes in genes associated with the spindle apparatus suggests that some of these changes may have been subject to positive selection in modern humans (*Hudson et al., 1987*). To test whether the eleven changes in the eight genes may have been beneficial, we applied a method to detect positive selection that occurred in the common ancestral population of modern humans after their split from archaic humans (*Peyrégne et al., 2017*). This method relies on a hidden Markov model to identify genomic segments where archaic human genomes fall outside of modern human variation, that is, where all modern humans share a common ancestor more recently than any common ancestor shared with archaic humans. Applying this approach to the genomes of a Neandertal (*Prüfer et al., 2014*), a Denisovan (*Meyer et al., 2012*) and 504 Africans (*1000 Genomes Project Consortium et al., 2015*), we identified such segments spanning 132–547,448 base pairs (bp) around the missense substitutions of each of the eight spindle genes (*Figure 1*; *Figure 1—figure supplement 1*). The genetic lengths of these segments are informative about how fast they spread in modern humans since the common ancestor shared with archaic humans. Segments longer than 0.025cM were not observed in neutral simulations (i.e. false positive rate <0.1%, *Peyrégne et al., 2017*) and therefore represent evidence for positive selection. We account for uncertainty in local recombination rates by using two recombination maps (African-American and deCODE maps, *Hinch et al., 2011*; *Halldorsson et al., 2019*). One gene, *SPAG5*, carries a segment around the missense substitutions longer than 0.025cM in both maps (*Figure 2*). The segments in two genes, *ATRX* and *KIF18A*, exceed this cutoff only for the African-American and deCODE recombination maps, respectively (*Figure 2A*). Thus, there is consistent evidence for positive selection on *SPAG5*, while evidence for positive selection affecting *ATRX* and *KIF18A* is dependent on the recombination maps used.

To gain further insights into the history of the missense substitutions in the spindle genes, we estimated the time when each of these mutations occurred. Note that this time differs from that of fixation, which may have happened much later, particularly in the absence of positive selection. As fixation events since the split with the ancestors of Neandertals and Denisovans are rare (*Peyrégne et al., 2017*), it is most parsimonious to assume that the mutations that reached fixation in the region of each spindle gene represent a single event of fixation (i.e. were linked to the missense substitution(s) as they spread in modern humans). Thus, estimates of the coalescence time of the haplotypes in which the substitutions are found today provides a most recent bound for when the mutations occurred. By contrast, the most recent common ancestors shared between modern and archaic humans (who carried the ancestral variants) will predate the occurrence of the mutations and therefore provide older bounds for the time of origin of the missense variants.

By computing pairwise differences among the high-quality genome sequences of 104 individuals from Africa (Human Genome Diversity Panel, *Bergström et al., 2020*; including two non-African individuals with the ancestral version of *KATNA1* and *KIF18A*, respectively; Methods), and four archaic human genomes (*Prüfer et al., 2017*; *Meyer et al., 2012*; *Prüfer et al., 2014*; *Mafessoni et al., 2020*), we estimated the ages of the missense substitutions of *KATNA1*, *KIF18A*, *KNL1*, *SPAG5*, and *STARD9* (*Figure 2B and C*, *Appendix 2—table 1*). We excluded *NEK6* and *RSPH1* as the regions identified around the missense substitutions were too short to estimate their age (4,104bp and 132bp, respectively). We also excluded *ATRX*, which is located on the X chromosome. The relative difference in age estimates suggests that the mutations in *KATNA1*, *KNL1,* and *STARD9* occurred much earlier than the mutations in *SPAG5* and *KIF18A* and may be more than a million years old. By contrast, the mutations in *SPAG5* and *KIF18A* are more recent. This is consistent with that the latter two genes have some evidence of positive selection around their missense substitutions whereas the former genes lack evidence for selection.

## A modern human-like *KNL1* haplotype in Neandertals

The identification of modern human-specific missense changes was based on the high-coverage genomes of one Neandertal and one Denisovan (*Prüfer et al., 2014*). With the availability of additional archaic human genome sequences, it is now possible to explore whether the derived states may have occurred in some archaic humans. For five of the spindle genes, sequence information from seven to ten archaic humans is available at the relevant positions (*Prüfer et al., 2017*; *Prüfer et al., 2014*; *Mafessoni et al., 2020*; *Hajdinjak et al., 2018*; *Peyrégne et al., 2019*; *Slon et al., 2018*). In none of them, there is evidence for the presence of the derived missense variants (*Appendix 3—table*

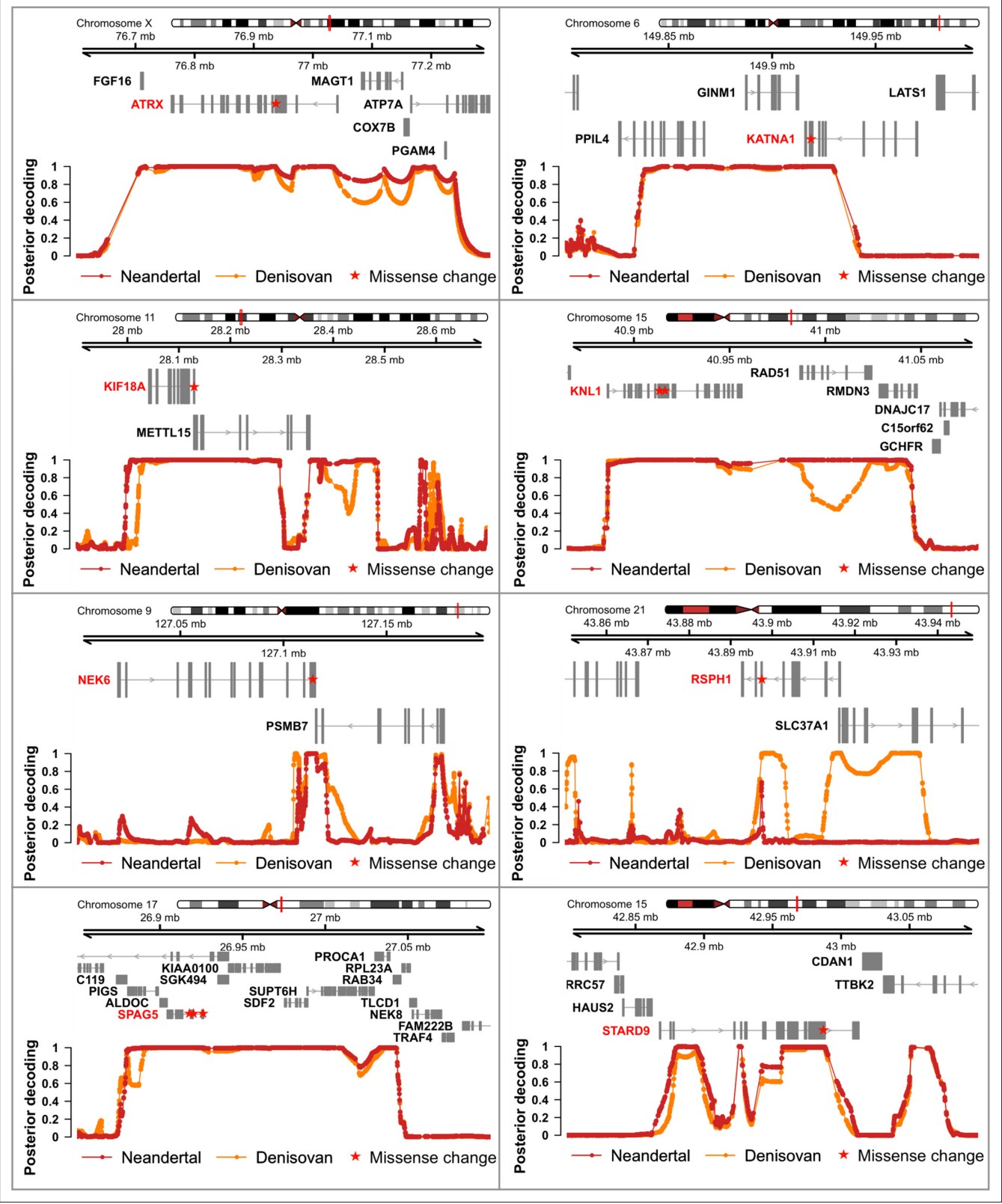

**Figure 1.** Genomic regions around spindle genes where archaic humans fall outside the modern human variation. Each panel corresponds to the region around the missense change(s) (red stars) in a spindle gene. The grey boxes correspond to exons. The curves give the posterior probability (computed as in *Peyrégne et al., 2017*) that an archaic genome (*Altai Neandertal* in red, *Denisova 3* in orange) is an outgroup to present-day African genomes at

*Figure 1 continued on next page*

a particular position (dots on the curves correspond to informative positions, that is polymorphic positions or fixed derived substitutions in Africans from the 1,000 Genomes Project phase III, compared to four ape genomes). Chromosomal locations are given on top.

The online version of this article includes the following figure supplement(s) for figure 1:

**Figure supplement 1.** Genomic regions where archaic humans fall outside the modern human variation, identified using the most recent deCODE recombination map (*Halldorsson et al., 2019*).

1). For one gene, *STARD9*, one DNA fragment sequenced from a Neandertal individual (*Denisova 5*, 52-fold average sequence coverage) carries the derived variant whereas 29 carry the ancestral variant. This is likely to represent present-day human DNA contamination or a sequencing error. For another gene, *SPAG5*, which carries three missense substitutions, two DNA fragments sequenced from a Neandertal individual (*Mezmaiskaya 1*, 1.9-fold average sequence coverage) carry the derived variant at one of these positions (chr17:26,919,034; hg19), whereas three DNA fragments carry the ancestral variant. No information was available for this individual at the two other positions, but none

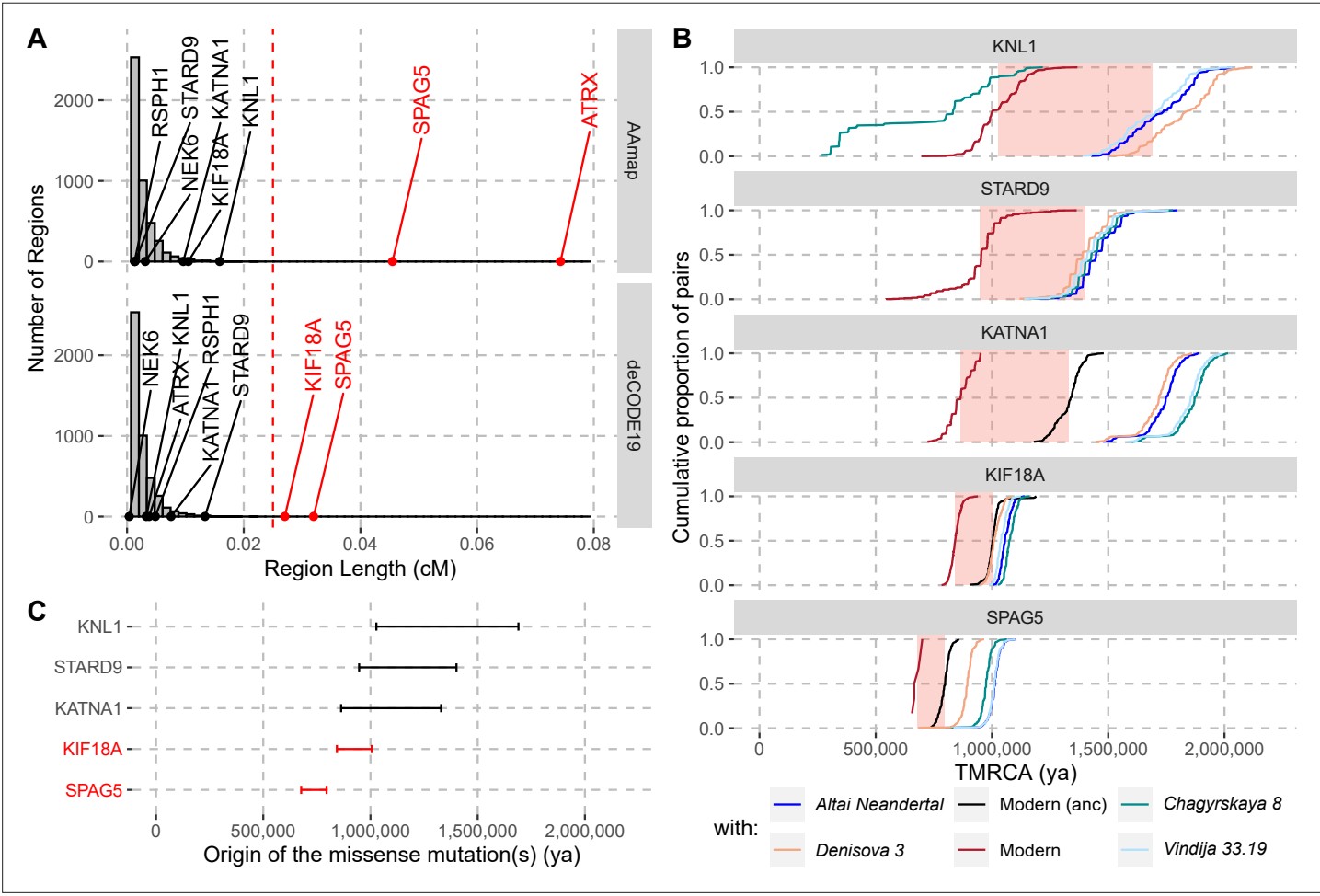

**Figure 2.** Evidence for selection in the spindle genes with age estimates of these substitutions. (**A**) The genetic length of segments around the missense substitutions where the Altai Neandertal and Denisova 3 fall outside the human variation (*Figure 1*) using the African-American map, AAmap, or the deCODE map, deCODE19. The grey histogram corresponds to the length distribution of such segments in neutral simulations (*Peyrégne et al., 2017*). Candidate genes for selection (red) are those with segments longer than 0.025cM (*Peyrégne et al., 2017*) (vertical red dashed line). (**B**) Cumulative distributions of pairwise times to the most recent common ancestors (TMRCA) among present-day African chromosomes with the most distant relationships (red; see Methods), or between the chromosomes of present-day Africans and either present-day individuals with the ancestral versions of the missense substitutions ('Modern (anc)', in black) or archaic humans (other colors). The pink areas correspond to estimated time intervals for the origin of the missense substitutions and their bounds correspond to the average TMRCAs over the red curve and the next one (back in time), respectively. (**C**) Summary of ages of substitutions as described in panel B. Genes with evidence of positive selection are highlighted in red.

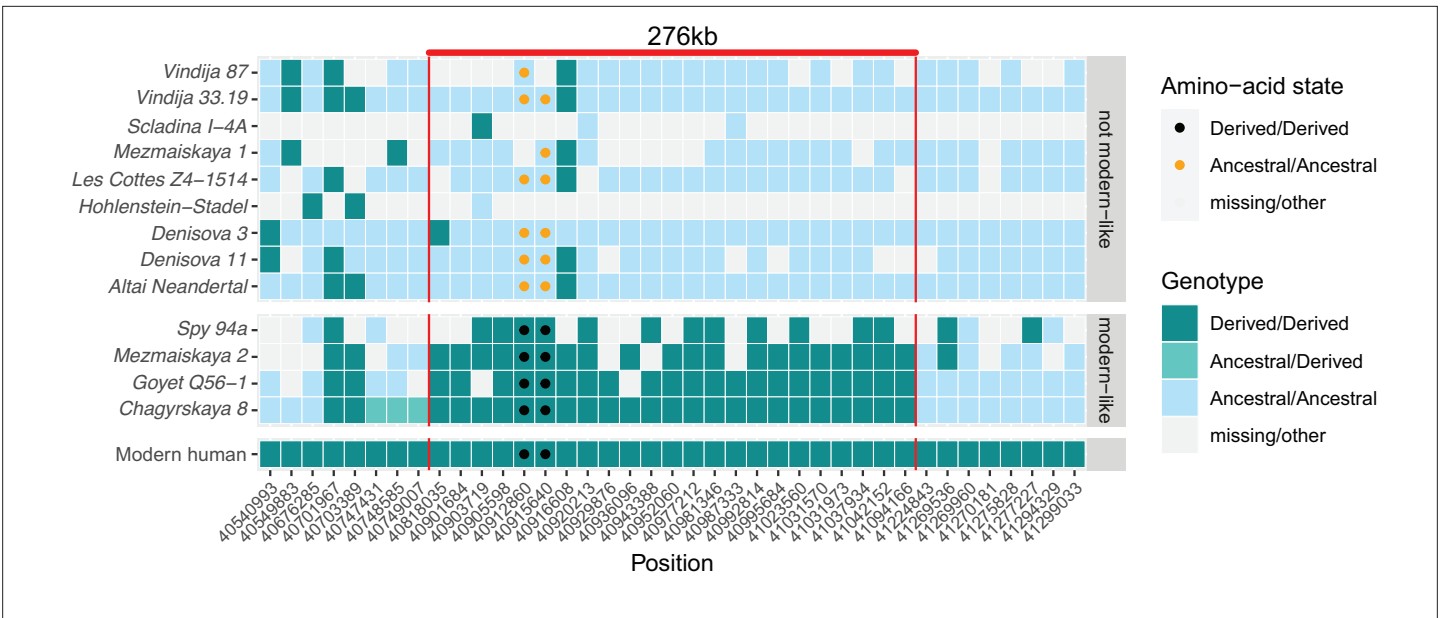

**Figure 3.** A modern human-like haplotype in some Neandertals. Genotypes from 13 archaic individuals (y-axis) are shown in a region around the two missense changes (dots) in *KNL1*. We only show positions (x-axis) that are derived in all Luhya and Yoruba individuals from the 1,000 Genomes Project compared to four great apes (*Peyrégne et al., 2017*) and at least one high-coverage archaic genome (*Chagyrskaya 8*, *Denisova 3*, *Vindija 33.19* and *Altai Neandertal*, i.e., *Denisova 5*). The colors of the squares and dots represent the genotype, with ancestral and derived alleles. For the low coverage archaic genomes, we randomly sampled a sequence at each position. Red lines indicate the modern human-like haplotype.

of eight neighbouring positions where present-day Africans carry fixed derived alleles and where information is available for this Neandertal carries any modern human-like allele (*Appendix 3—table 2*). This could therefore represent present-day human DNA contamination, which amounts to 2–3% in the DNA libraries sequenced from this individual (*Prüfer et al., 2017*).

In contrast, between one and 27 DNA fragments that cover the two missense substitutions in *KNL1* in four of the twelve Neandertals available carried only derived alleles (*Figure 3*). This includes the *Chagyrskaya 8* genome, which is sequenced to 27-fold average genomic coverage. Several of the fragments carrying derived alleles also carry cytosine-to-thymine substitutions near their ends, which is typical of ancient DNA molecules and suggests that the fragments do not represent present-day human DNA contamination (*Briggs et al., 2007*).

Further evidence that the modern human-like alleles in *KNL1* in the four Neandertals are not due to present-day DNA contamination comes from the observation that they carry modern human-like derived alleles that occur in all present-day Africans at 21 other positions in a 276kb-long region around the missense variants, whereas the other Neandertal and Denisovan individuals who carry the ancestral alleles do not (*Figure 3*). In addition, as there is no evidence of ancestral alleles at any of these informative positions, the four Neandertals probably all carried this 'modern human-like' haplotype in homozygous form.

To investigate how the divergence among Neandertals of the haplotypes with and without the derived missense substitutions in the *KNL1* gene compares to other regions of Neandertal genomes, we calculated the divergence in non-overlapping 276kb-segments between the *Altai Neandertal* (*Denisova 5*), who carries ancestral *KNL1* substitutions, and *Chagyrskaya 8*, who carries the derived substitutions (*Figure 4A*). The number of differences in the *KNL1* region is about one order of magnitude higher than the average across the genome and in the top 0.27% of all regions in the genome. Thus, the *KNL1* region stands out as unusually diverged between these two Neandertals.

## Introgression of the *KNL1* haplotype into Neandertals

It is intriguing that the Neandertals who carry the two missense changes in *KNL1* also carry modern human-like alleles shared by all (or nearly all) present-day Africans at multiple positions in the region of *KNL1* and exhibit unusually high divergence to other Neandertal haplotypes. This raises the

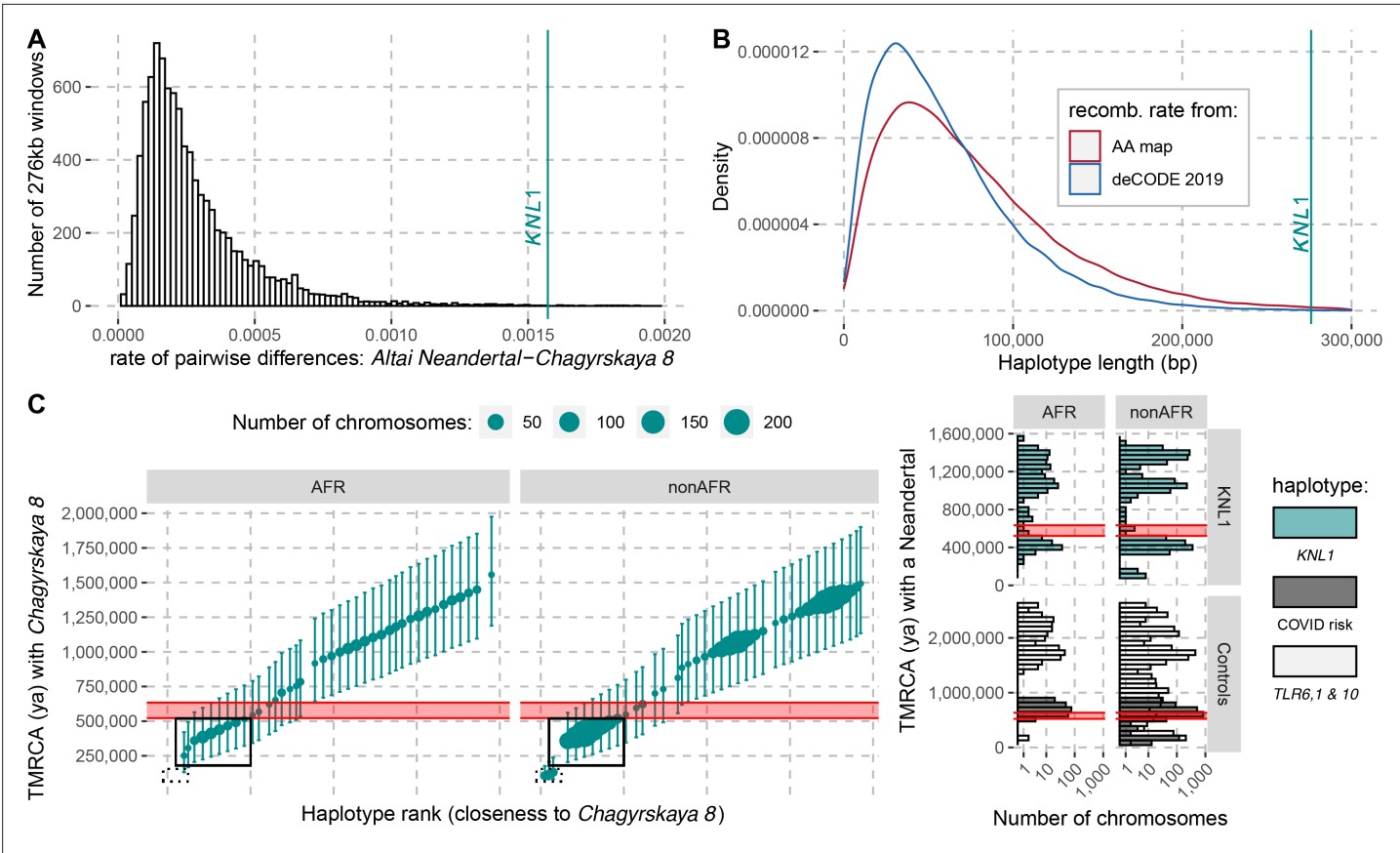

**Figure 4.** The modern human-like KNL1 haplotype in Neandertals. (**A**) Pairwise differences between two high coverage Neandertal genomes (*Chagyrskaya 8* and *Altai Neandertal* (*Denisova 5*)) in non-overlapping sliding windows of 276 kb (histogram) and in the *KNL1* region (vertical cyan line; chr15:40,818,035–41,094,166, hg19). Windows with less than 10,000 genotype calls for both Neandertals were discarded. (**B**) The expected length distributions under a model of incomplete lineage sorting based on local recombination rate estimates from the African-American (AA) and deCODE recombination maps and the length of the modern human-like *KNL1* haplotype in Neandertals (vertical cyan line). (**C**) Left panel: Time to the most recent common ancestor (TMRCA) between the *Chagyrskaya 8* Neandertal (who carries the modern human-like haplotype) and *KNL1* haplotypes in present-day humans with their 95% confidence intervals (bars) for chr15:40,885,107–40,963,160 (hg19). The size of the points corresponds to the number of chromosomes carrying this haplotype in the HGDP dataset. The black rectangles highlight subsets of haplotypes with TMRCAs more recent than the modern-archaic population split time (***Prüfer et al., 2017***) (shaded pink area). Right panel: Distribution of pairwise TMRCAs between the Neandertal and present-day humans from the HGDP dataset in the region of *KNL1* and two other regions with archaic haplotypes in present-day humans (Controls, ***Zeberg and Pääbo, 2020***; ***Dannemann et al., 2016***; COVID risk region: chr3:45,859,651–45,909,024; *TLR6, 1 & 10*: chr4:38,760,338–38,846,385). We used the *Vindija 33.19* genome for the COVID risk haplotype and the *Chagyrskaya 8* genome otherwise.

The online version of this article includes the following figure supplement(s) for figure 4:

**Figure supplement 1.** Genotypes of the 12 non-African individuals that inherited one copy of *KNL1* from archaic humans.

question if Neandertals may have inherited this haplotype from ancestors or relatives of modern humans. However, the age of the *KNL1* missense mutations predates the divergence of Neandertals and modern humans (***Figure 2C***) and must thus have been segregating in the ancestral populations of the two groups. This may have resulted in the presence of the derived variants in both Neandertals and modern humans even in the absence of any gene flow ('incomplete lineage sorting'). However, in this case, the segment carrying similarity between Neandertal and modern human genomes is expected to be shorter than if it entered the Neandertal population by gene flow because the number of generations over which recombination would have had the opportunity to shorten the haplotype would be larger.

We used local estimates of the recombination rate (Methods) and a published method (***Huerta-Sánchez et al., 2014***) to infer the expected length distribution for haplotypes inherited from the population ancestral to Neandertals and modern humans. The 276kb haplotype is longer than expected (***Figure 4B*** and $p \leq 0.006$) and it is therefore unlikely to be inherited from the common ancestral

population. Thus, although the missense variants were segregating in the common ancestral population of Neandertals and modern humans, Neandertals did not inherit this haplotype from that population. Rather, the haplotype in Neandertals is likely the result of gene flow between Neandertals and modern humans. Since all present-day humans carry the derived *KNL1* variants whereas only some Neandertals do, and since the modern human-like *KNL1* haplotype is unusually diverged to other Neandertal haplotypes, gene flow was likely from modern humans (or their relatives) to Neandertals.

## Age of *KNL1* gene flow into Neandertals

Using the length of the haplotype, the local recombination rate (averaged over the African-American and deCODE recombination maps), the age of the most recent Neandertal carrying the haplotype (~40kya, *Hajdinjak et al., 2018*), and given the limitation that recombination among the fixed alleles in modern humans cannot be detected, we estimate an older age limit for the haplotypes seen in Neandertals and modern humans of 265kya (Methods). This means that gene flow occurred after this time.

It is possible to further refine the age estimate of this gene flow if some present-day humans are closer to the Neandertals carrying the haplotype than other present-day humans. By comparing the high-quality genomes of Neandertals with and without the modern-like *KNL1* haplotype, we identified 206 positions that are derived on the modern-like *KNL1* haplotype in Neandertals. We then computed the frequency of these alleles among 104 present-day African genomes and identified 19 positions where 32–39% of present-day people share the derived allele (*Appendix 4—table 1*). These derived alleles are physically linked ($r^2$ >0.58) and tag a 78kb-long haplotype (102kb in some individuals, $r^2$ >0.39) where some present-day individuals are closer to the *Chagyrskaya 8* Neandertal than other present-day individuals. The individuals with the 102kb-long haplotype carry seven differences to the *Chagyrskaya 8* Neandertal (among the 36,106 bases called). This number of differences yields an estimate for the most recent common ancestor with Neandertals carrying the modern-like *KNL1* haplotype of 251kya (95% CI: 131-421kya; *Figure 4C*). Note that this haplotype is also too long to be inherited from the common ancestral population of Neandertals and modern humans half a million years ago (95% CI for the age of the last common ancestor: 49-304kya). The length of the 102kb-long haplotype in combination with the number of differences to *Chagyrskaya 8* yields an age estimate of 202kya (95% CI: 118-317kya, Methods), consistent with the estimate of gene flow <265kya based on the haplotype length in Neandertals.

In conclusion, some Neandertals carry a haplotype encompassing the *KNL1* gene that carries derived alleles also seen in all present-day humans. The length of this haplotype suggests that it entered the Neandertal population less than 265kya. Some present-day people carry a haplotype in this region that share a common ancestor with the modern human-like haplotype in Neandertals 202kya. Both estimates represent times after which the contact that contributed this haplotype to Neandertals must have occurred.

## No evidence for selection on the *KNL1* haplotype in Neandertals

The oldest Neandertal carrying the modern human-like *KNL1* haplotype for whom we have genome sequence data is *Chagyrskaya 8*, who lived 60-80kya in the Altai Mountains (*Mafessoni et al., 2020*). It is not present in two older Neandertals from Europe (*Peyrégne et al., 2019*) and one from Siberia (*Prüfer et al., 2014*), nor in two other archaic individuals who lived around 60-80kya in Siberia and the Caucasus (*Prüfer et al., 2017*; *Slon et al., 2018*). However, it is present in three out of five Neandertals who lived 40-50kya (*Prüfer et al., 2017*; *Hajdinjak et al., 2018*) in Europe, which may hint at an increase in its frequency over time.

To test this hypothesis, we investigated how often frequencies of other variants (not necessarily introgressed) in the genomes of these Neandertals similarly increased over time. We identified 7,881 polymorphic transversions at least 50kb apart where the derived allele was absent among the three early Neandertals but present at least once in the Neandertals who lived 60-80kya and 40-50kya. Among these positions, at least three late Neandertals carried the derived alleles at 2,773 positions (35.2%), suggesting that such changes in frequency were common. Although this analysis is limited by the few Neandertal genomes available, it yields no evidence for positive selection on the *KNL1* variants in Neandertals.

### Reintroduction of the modern-like *KNL1* haplotype into non-Africans

As several late Neandertals carried a modern human-like *KNL1* haplotype, it is possible that it was reintroduced into modern humans when they met and mixed outside Africa approximately 44-54kya (*Iasi et al., 2021*). We would then expect that some non-Africans would carry a haplotype that is more closely related to that of *Chagyrskaya 8* than the present-day humans analyzed above (those highlighted by a solid box in *Figure 4C*). Among 825 non-African individuals from around the world, we identified 12 individuals from several populations that carry one chromosome that differs at just 7–13 positions from the *Chagyrskaya 8* genome in the *KNL1* region (out of ~140kb with genotype calls in the 276kb region; *Appendix 5—table 1*). By comparison, other non-African individuals carry 54–179 differences to the *Chagyrskaya 8* genome in this region (*Figure 4C*). We estimated that the 12 individuals share a most recent common ancestor with *Chagyrskaya 8* about 96 to 139kya in this region of *KNL1* (95% CI: 81-145kya to 95-198kya for the individuals with 7 and 13 differences to *Chagyrskaya 8*, respectively; Methods). These estimates are similar to those computed in Neandertal haplotypes previously described in present-day individuals (*Figure 4C*, right panel, *Zeberg and Pääbo, 2020*; *Dannemann et al., 2016*). Adjoining the 3'-end of this *KNL1* region, there are seven positions where these 12 individuals share alleles with archaic genomes (including four alleles shared with *Chagyrskaya 8*; *Figure 4—figure supplement 1*) while no other present-day humans in the dataset do so. These observations suggest that these 12 present-day individuals carry a *KNL1* haplotype inherited from Neandertals.

Although the missense alleles in *KNL1* are fixed among the genomes of 2,504 present-day humans (*1000 Genomes Project Consortium et al., 2015*), some rare ancestral alleles may exist among present-day human genomes, for example, due to interbreeding with archaic humans or due to back mutations. We therefore looked at the exome and whole-genome sequences from the *gnomAD* database (v2.1.1, *Karczewski et al., 2020*). There are no ancestral alleles among the exome sequences (out of 227,420 alleles called). One of the 15,684 whole genomes carries one ancestral allele at one of the two positions carrying missense variants. This may be a back mutation (rs755472529; *Appendix 6—table 1*). Thus, although there is evidence that gene flow from Neandertals introduced the derived version of *KNL1* into modern humans, there is no evidence of the ancestral version of *KNL1*, which was also present in Neandertals, in present-day people. This is compatible with that the ancestral variants may have been disadvantageous and therefore eliminated after admixture with Neandertals and Denisovans.

## Discussion

Spindle genes experienced an unusually large number of missense changes in modern humans since the split from a common ancestor with archaic humans. This is intriguing, as mitotic metaphase has been shown to be prolonged in apical progenitors during human brain organoid development when compared to apes (*Mora-Bermúdez et al., 2016*; *Mora-Bermúdez et al., 2021*). Some of the missense changes in spindle genes that occurred since the separation of modern and archaic humans may be involved in such differences.

One of the spindle genes carrying missense changes, *KNL1*, has a particularly interesting evolutionary history in that some Neandertals carried a haplotype sharing two missense variants with present-day humans that were hitherto believed to be specific to modern humans. Both the length of the modern-like *KNL1* haplotype in Neandertals and the small number of differences between present-day humans and Neandertals in this region suggest that they shared a common ancestor more recently than the estimated split time between these two populations. While only a few Neandertals carry this *KNL1* haplotype and its divergence to other Neandertal haplotypes is unusually high (*Figure 4A*), many variants on the haplotype are fixed or common among present-day humans. Therefore, we infer that ancestors (or relatives) of modern humans contributed this haplotype to Neandertals. *Figure 5* summarizes the complex history of *KNL1*.

That groups related to early modern humans contributed a *KNL1* haplotype to early Neandertals supports previous evidence based on the analyses of mitochondrial DNA (*Meyer et al., 2016*; *Posth et al., 2017*), Y chromosomes (*Petr et al., 2020*) as well as genome-wide data (*Kuhlwilm et al., 2016*; *Hubisz et al., 2020*) for early contacts between populations related to modern humans and Neandertals in Eurasia. The estimated ages of the most recent common ancestor of the *KNL1* haplotype

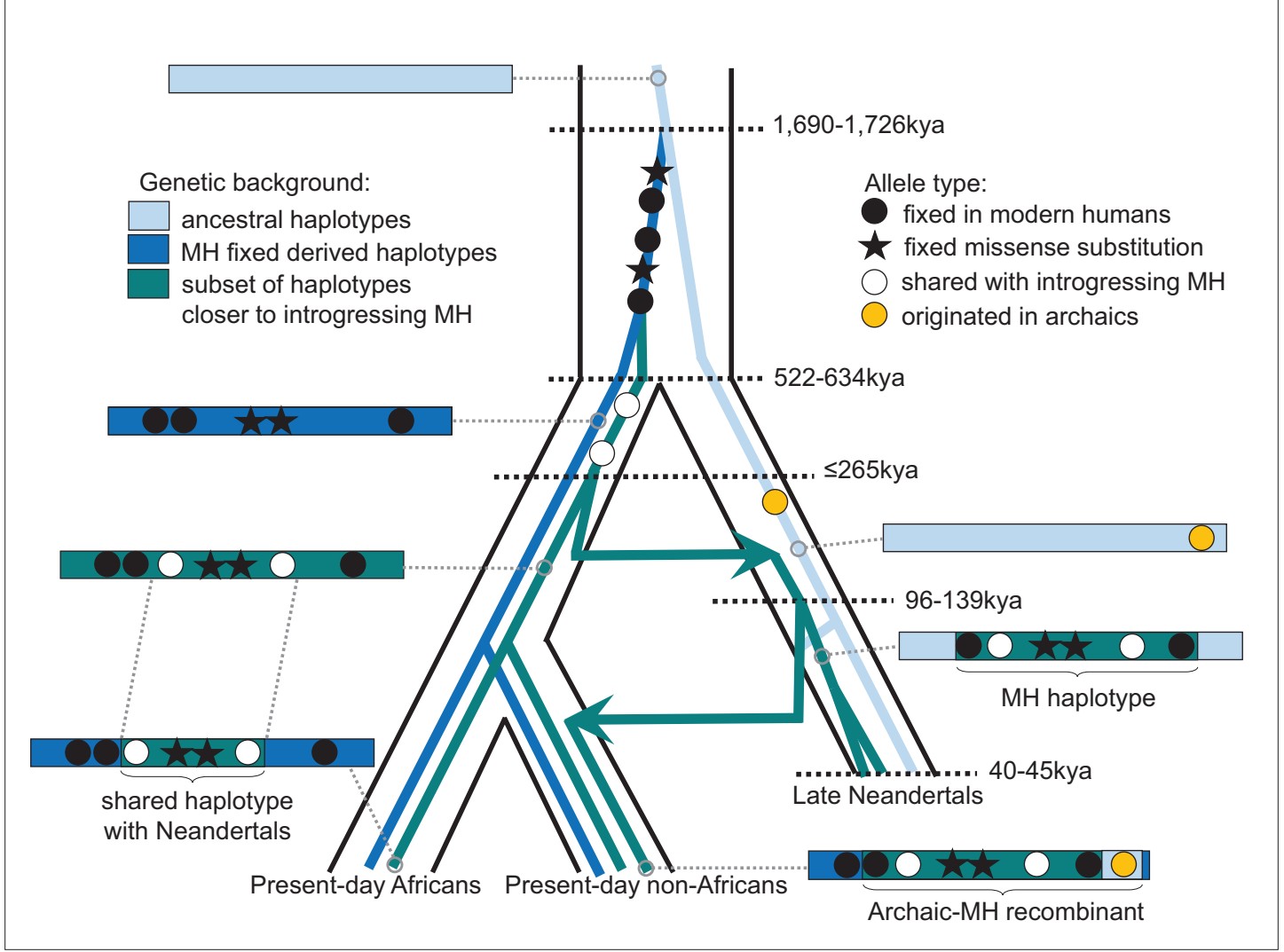

**Figure 5.** Schematic illustration of the history of KNL1. The tree delineated in black corresponds to the average relationship between the modern and archaic human populations. The inner colored trees correspond to the relationship of different *KNL1* lineages, with arrows highlighting the direction of gene flow between populations. The corresponding haplotypes are illustrated on the sides of the tree and show the recombination history in the region (e.g. the recombinant Neandertal haplotype with variants of putative archaic origin in non-Africans). Dots correspond to informative positions, and the stars illustrate the missense substitutions. The age of relevant ancestors are marked by horizontal dotted lines. MH: Modern human.

of ~202kya (based on pairwise differences and length of the shared haplotype) and ≤265kya (based on the length of the haplotype in Neandertals) suggest that at least some contacts were more recent than 265kya and are in agreement with some of the previous age estimates for such contacts (*Hubisz et al., 2020*). This could resolve the apparent discrepancy between the high mitochondrial diversity relative to their nuclear diversity among early Neandertals (*Peyrégne et al., 2019*) as the mitochondrial diversity would reflect diversity inherited from a potentially more diverse modern human population. Notably, human remains that may be morphologically similar to modern humans and are of a relevant age have been found in Greece ~210kya (*Harvati et al., 2019*) and in the Levant 177–194kya (*Hershkovitz et al., 2018*). In any event, the complex history of the *KNL1* haplotype illustrates the close and intricate relationship between Neandertals and modern humans.

The enrichment of missense changes in spindle genes (*Prüfer et al., 2014*; *Pääbo, 2014*) suggests that the spindle apparatus may have played a special role during modern human evolution. We show evidence of positive selection in the region around the missense variants in *SPAG5*, and perhaps in *ATRX* and *KIF18A*. Although there is no evidence of selection for the other variants it should be kept in mind that, based on simulations, the method used only detects selective advantages of at least

0.5% with a 65% probability (*Peyrégne et al., 2017*). It is also unlikely to detect selection starting from standing variation. Therefore, the absence of evidence for selection does not exclude the possibility of selection. For instance, it is interesting that only derived *KNL1* haplotypes introduced into modern humans from Neandertals persisted until present-day. This provides suggestive evidence that the missense mutations in *KNL1* may also have been important for modern humans. Ultimately, functional characterization of the potential effects of the ancestral and derived variants in spindle genes is necessary to clarify if these changes had effects that could have been advantageous.

### Note added in proof

We refer readers to *Mora-Bermúdez et al., 2022* who show an effect of the missense changes in *KIF18A* and *KNL1* on the prolongation of metaphase.

## Methods

### Identifying genes associated with the spindle apparatus

The human Gene Ontology Annotation database was downloaded on June 8th 2021 (goa_human.gaf. gz, from http://current.geneontology.org/products/pages/downloads.html), together with the basic Gene Ontology terms (http://purl.obolibrary.org/obo/go/go-basic.obo). We selected all the Gene Ontology terms that include the keyword 'spindle' (65 terms), identified all the human genes that are associated with at least one of these terms (562 genes out of 19,719) and overlapped them with the list of 87 genes that exhibit fixed missense changes in modern humans (*Prüfer et al., 2014*; *Pääbo, 2014*).

### Testing for enrichment of missense changes in spindle genes

The dbNSFP (version 4.2), a database of all potential missense variants in the human genome, was downloaded on October 4th 2021 from ftp://dbnsfp@dbnsfp.softgenetics.com/dbNSFP4.2a.zip (*Liu et al., 2020*). After removing the variants that do not have a coordinate in hg19 and do not pass the minimal filters for the *Altai Neandertal* (*Denisova 5*) and Denisovan (*Denisova 3*) genomes (*Prüfer et al., 2017*), we randomly sampled 96 of the missense variants reported in this database (from the 62,961,368 remaining after filtering) a thousand times and recorded each time how many variants belonged to a spindle gene (as identified in the previous section). If a variant belonged to multiple genes, it was counted only once. The p-value reported in the Results section corresponds to the number of repetitions with eleven or more missense changes in spindle genes. Similarly, to test whether multiple missense changes in a single gene are expected by chance, we counted how often among the thousand repetitions there was at least one gene with two (or three) or more missense changes. Finally, to test whether eight spindle genes with missense changes is expected by chance, we instead counted how often there were eight or more spindle genes with at least one missense change in these resamples.

### Processing the human genome datasets and great ape reference genomes

For most analyses, we used phased genotypes from the high-coverage genomes from the Human Genome Diversity Panel (HGDP, *Bergström et al., 2020*), as well as gVCF files for each individual to identify positions which have genotype calls. After subsetting to regions of interest (*Appendix 2—table 1*), gVCF files were converted into VCF files, concatenated with the phased genotypes and merged into a single VCF file using bcftools (*Danecek et al., 2021*). Genotypes that did not pass all the quality filters from *Bergström et al., 2020* were then discarded. Finally, positions were lifted over to hg19 using picard tools (default parameters, http://broadinstitute.github.io/picard) and the chain file hg38ToHg19 from the University of California, Santa Cruz, (UCSC) Genome Browser (hgdownload. cse.ucsc.edu/goldenPath/hg38/liftOver/hg38ToHg19.over.chain.gz).

Genotypes from four high-coverage archaic genomes (*Denisova 3* (*Meyer et al., 2012*), *Altai Neandertal* (*Prüfer et al., 2014*), *Vindija 33.19* (*Prüfer et al., 2017*) and *Chagyrskaya 8* (*Mafessoni et al., 2020*), as generated in *Prüfer et al., 2017* and *Mafessoni et al., 2020* with snpAD [*Prüfer, 2018*]) and nine low-coverage archaic genomes (*Goyet Q56-1*, *Mezmaiskaya 2*, *Les Cottés Z4-1514*, *Vindija 87* and *Spy 94a* from *Hajdinjak et al., 2018*, *Hohlenstein-Stadel* and *Scladina I-4A* from *Peyrégne*

*et al., 2019* and *Mezmaiskaya 1 Prüfer et al., 2017*) were merged into a single VCF file. For the low-coverage archaic genomes, one base with a quality of at least 30 was sampled randomly at each position. Positions outside the mappability track 'map35_100' (i.e. positions outside regions where all overlapping 35-kmers are unique in the genome, *Prüfer et al., 2017*) were filtered out, as well as positions without at least one genotype call from the high-coverage archaic genomes. Note that we also checked the bases carried by all sequences overlapping the substitutions in spindle genes from the published BAM files of these archaic human genomes using samtools tview (*Danecek et al., 2021*).

For each of these datasets, ancestral allele information was retrieved from great ape reference genome assemblies: chimpanzee (panTro4) (*Chimpanzee and Analysis, 2005*), bonobo (panPan1.1) (*Prüfer et al., 2012*), gorilla (gorGor3) (*Scally et al., 2012*), and orangutan (ponAbe2) (*Locke et al., 2011*) as LASTZ alignments to the human genome GRCh37/hg19 prepared in-house and by the UCSC Genome Browser (*Harris, 2007*). We defined the ancestral allele as the one carried by at least three of the four ape reference genome assemblies, allowing a third allele or missing information in only one ape reference genome.

## Processing the recombination maps

To compute genetic distances, we used a recombination map obtained from crossovers between African and European ancestry tracts in African-Americans (*Hinch et al., 2011*, available in hg19 coordinates from http://www.well.ox.ac.uk/~anjali/) and a map based on crossovers in parent-offspring pairs from Iceland (deCODE, *Halldorsson et al., 2019*). The latter was lifted over to hg19 (originally in hg38 coordinates) with the liftover tool and the chain file hg38ToHg19 from UCSC (using a minimum match rate of 0.9 between bases of both assemblies). This resulted in both gaps and overlaps between windows of the recombination map. Therefore, we assumed that the recombination rate in a gap was the average of the two directly adjacent windows, and we truncated the windows that overlap with a previous window (or removed the window if it overlapped completely with the previous one).

## Investigating signals of selection in modern humans

For the analysis of positive selection in modern humans, we used bi-allelic single nucleotide polymorphisms (SNPs) from 504 African individuals from the 1,000 Genomes Project phase III (*1000 Genomes Project Consortium et al., 2015*), excluding African populations that may have European ancestry, that is, African Caribbeans in Barbados (ACB) and individuals with African Ancestry in Southwest US (ASW). We also compiled a list of sites where Africans differ from the common ancestor with chimpanzee. These are positions that are not polymorphic among the 504 African individuals and where six high-coverage African genomes (Mbuti, San and Yoruban individuals from *Prüfer et al., 2014*) were identical but differed from four great ape reference genome assemblies (panTro4, panPan1.1, gorGor3, and ponAbe2). We then extracted the genotypes of the *Altai Neandertal* (*Denisova 5*) or Denisovan (*Denisova 3*) genomes at these positions, and only considered those that pass the minimal filters for each genome, respectively (*Prüfer et al., 2017*). Note that we used the genotypes called with snpAD (*Prüfer, 2018*) and generated in the same study as the minimal filters (*Prüfer et al., 2017*).

We tested whether the missense substitutions in the spindle genes overlap regions displaying signatures of ancient selective sweeps using the hidden Markov model described in *Peyrégne et al., 2017*. We executed that method independently for each chromosome and for both the *Altai Neandertal* and *Denisova 3* genomes, using genetic distances computed from the African-American or deCODE recombination maps (*Hinch et al., 2011*; *Halldorsson et al., 2019*). We then identified regions around the missense substitutions of spindle genes where the archaic genome fall outside the human variation ('external' regions) with a posterior probability above 0.2. We applied this cutoff on the sum of the posterior probabilities of both 'external' states (i.e. generated either from drift or from a selective sweep, *Peyrégne et al., 2017*). We further intersected the regions called with the *Altai Neandertal* and *Denisova 3* genomes and measured the genetic length of the overlap to determine whether there is evidence for selection.

## Reconstructing the chronology of the missense substitutions

To get an upper age estimate of the missense substitutions, we estimated the TMRCA between Africans and archaic humans in the regions around the missense substitutions where archaic humans fall outside the human variation. We computed the number of pairwise differences between each African

chromosome and each high-coverage archaic human genome (*Prüfer et al., 2017*; *Meyer et al., 2012*; *Prüfer et al., 2014*; *Bergström et al., 2020*; *Mafessoni et al., 2020*) sampling a random allele at heterozygous positions in the archaic genomes. The age of the common ancestor of two individuals conditional on the number D of pairwise differences follows a gamma distribution with shape parameter $\alpha$=D + 1 and rate parameter $\beta$=$\theta$N+1 (56), where $\theta$ is the population scaled mutation rate (i.e. $4N_e\mu$ with $\mu$ and $N_e$ denoting the mutation rate and the effective population size, respectively) and N is the number of bases with genotype calls in both individuals. This model assumes that the prior probability of coalescence follows an exponential distribution, with rate equal to 1 when time is scaled in units of the diploid effective population size ($2N_e$), and that both individuals are sampled in the present. To account for a branch shortening S associated with the age of ancient individuals, we truncated the posterior distribution so that the age of the common ancestor cannot be more recent than S and we added $\frac{S}{2}$ as a correction for this branch shortening. The expected TMRCA is then:

$$E\left(T|T \geq S\right) = \frac{1}{\beta}\left(\alpha + \frac{\left(\frac{\beta S}{2}\right)^{\alpha}e^{-\frac{\beta S}{2}}}{\Gamma\left(\alpha, \frac{\beta S}{2}\right)}\right) + \frac{S}{2}$$ (Equation 1, Appendix 7), with $\Gamma\left(\alpha, \frac{\beta S}{2}\right)$ denoting the

upper incomplete gamma function with lower limit $\frac{\beta S}{2}$, which we computed with the gammainc() function from the R library expint v0.1–6. We assumed $\mu$ is $1.45 \times 10^{-8}$ mutations per base pair per generation (generation time of 29 years, *Scally and Durbin, 2012*) and a branch shortening of 50ky for *Vindija 33.19* (*Prüfer et al., 2017*), 70ky for *Denisova 3* (*Prüfer et al., 2017*), 80ky for *Chagyrkaya 8* (*Mafessoni et al., 2020*), and 120ky for the *Altai Neandertal* (*Denisova 5*) (*Prüfer et al., 2017*). We computed the 95% confidence intervals with the qtgamma() function of the R library TruncatedDistributions v1.0. Note that we also estimated the TMRCA in these genomic regions between Africans and a Papuan (HGDP00548), a Sindhi (HGDP00163) or a Biaka (HGDP00461) because they carry the ancestral versions of the missense substitutions in *KATNA1*, *KIF18A*, and *SPAG5*, respectively, and constrain further the upper age estimates of the missense substitutions.

To get a lower age estimate of the missense substitutions, we computed the number of pairwise differences among African chromosomes carrying the derived versions of the missense substitutions from the HGDP dataset (all African individuals, except HGDP00461 for *SPAG5*). For each region, we identified the two chromosomes that are the most distantly related (with the highest number of differences) and classified all the other African chromosomes into two groups depending on which of the two former chromosomes they are the most closely related in this genomic region. A chromosome was assigned to a group if it shared at least two derived alleles with the African chromosome used for defining this group but no more than one shared derived allele with the African chromosome defining the other group. This removes potential recombinants that could bias downward the estimate of the TMRCA. For each individual, we then counted the number of pairwise differences with individuals from the other group, restricting this analysis to positions that are genotyped in the high-coverage archaic genomes. Finally, we converted these counts into estimates of the TMRCA using the same equation as above, albeit with no branch shortening (i.e. $E\left(T\right) = \frac{\alpha}{\beta}$).

## Testing the hypothesis of gene flow in the *KNL1* gene

To test whether the *KNL1* haplotype identified in Neandertals could be shared with modern humans because of incomplete lineage sorting, we computed the probability that a haplotype shared since the common ancestral population is as long as, or longer than, the haplotype identified in *KNL1*. This approach was previously applied to the *EPAS1* haplotype shared between Tibetans and Denisovans (*Huerta-Sánchez et al., 2014*), but we briefly describe it here for completeness. The expected length L for a shared sequence is 1/(r x T), denoting r and T as the recombination rate and the length of the Neandertal and modern human branches since divergence (in generations of 29 years), respectively. As the ancestors of Neandertals and modern humans split from each other around 550kya (*Prüfer et al., 2017*), the most recent common ancestor shared between modern humans and Neandertals cannot be younger than this split time, in the absence of gene flow. As the modern-like *KNL1* haplotype is present in Neandertals that lived about 40kya (*Hajdinjak et al., 2018*), we set T=550,000–40,000=510,000 years, which corresponds to 17,586 generations of 29 years. Note that we do not include here the length of the modern human branch to be conservative, as we do not know when the substitutions that define the haplotype reached fixation in modern humans. Relying on local recombination rate estimates from the African-American map (0.148cM/Mb, *Hinch et al., 2011*) and the deCODE map (0.191 cM/Mb, *Halldorsson et al., 2019*), the expected length L of a haplotype shared

through ILS (i.e. 1/(r x T)) is 29.8kb or 38.5kb depending on the recombination map used. Assuming that the length of a shared sequence follows a Gamma distribution with shape parameter 2 and rate parameter 1 /L, the probability that such a sequence is as long as or longer than 276kb is then 1 − GammaCDF(276000, shape = 2, rate = 1 /L). This probability is ≤0.006 and hence a model without gene flow is rejected.

## Estimating the time of gene flow

To estimate the time of gene flow, we used an analogous model to that described above (*Albers and McVean, 2020*, adapted here to account for the branch shortening associated with the age of ancient individuals). In this model, the age of the common ancestor of two individuals conditional on the length L of their shared haplotype follows a gamma distribution with shape parameter $\alpha$=3 and rate parameter $\beta$=2$\rho$L+1, where $\rho$ is the population scaled recombination rate (i.e. $4N_er$ with r and $N_e$ denoting the recombination rate and the effective population size, respectively). To account for a branch shortening S, we applied again *Equation 1*, assuming that S is 40,000 years. In the case of the 276kb haplotype in Neandertals that carries alleles that are fixed in present-day humans, we made the conservative assumption that recombinations could not be observed in modern humans (i.e. the alleles were already fixed at the time of introgression) and, therefore, multiplied the age estimates by 2. We did not apply this correction for the estimate based on the 102kb haplotype shared between some present-day humans and Neandertals, as the length of this haplotype depends on the number of recombination events in both Neandertals and modern humans. Using the average recombination rate over the African-American and deCODE recombination maps (0.169cM/Mb), the expected age is 131kya based on the 276kb haplotype in Neandertals and 138kya based on the 102kb shared haplotype. We computed the 95% confidence intervals (83-265kya for the 276kb haplotype and 49-304kya for the 102kb haplotype) with the qtgamma() function of the R library TruncatedDistributions v1.0.

As another estimate for the time of gene flow, we look at the genetic distance between modern humans and Neandertals. For this purpose, we estimated the TMRCAs between *Chagyrskaya 8* (the only high coverage genome with this haplotype) and each present-day human genome from the HGDP dataset in the region chr15:40,885,107–40,963,160 (hg19). We randomly sampled one allele at heterozygous positions where the phase is unknown in these genomes (i.e., every heterozygous position of *Chagyrskaya 8*). The TMRCA conditional on the number D of pairwise differences follows a gamma distribution with parameters $\alpha$=D + 1 and $\beta$=$\theta$N+1 (see dating analysis of the missense substitutions). However, conditional on both the observed divergence and the length of the shared haplotype, the TMRCA follows a gamma distribution with shape parameter $\alpha$=D + 3 and rate parameter $\beta$=$\theta$N+2$\rho$L+1. In both cases, we accounted for the branch shortening of *Chagyrskaya 8* as described above (*Equation 1*, S=80,000 years) and assumed $1.45 \times 10^{-8}$ mutations per base pair per generation (generation of 29 years). The 95% confidence intervals were computed as described above.

As these analyses might be sensitive to the value of μ, we tested whether μ in the region of *KNL1* may differ from the genome-wide average by computing local estimates of the mutation rate in family trios from Iceland (*Jónsson et al., 2017*). We counted the number of de novo mutations in the genomes of the probands and divided this number by twice the length of the region and the number of trios in this dataset (1,548) to get a mutation rate estimate. This estimate was similar to the genome-wide average (Appendix 8).

## Testing whether the modern-like *KNL1* haplotype was under selection in Neandertals

To test whether the increase of the modern-like *KNL1* haplotype frequency in Neandertals is unexpected, we quantified how often positions in the genome exhibit similar frequency changes. We identified positions where early Neandertals (the *Hohlenstein-Stadel*, *Scladina I-4A* and *Denisova 5* Neandertals) carry the ape-like allele, whereas at least one individual carries the derived allele both among those that lived 60-80kya (*Chagyrskaya 8, Mezmaiskaya 1, Denisova 11*) and among those that lived 40-50kya (*Vindija 33.19, Goyet Q56-1, Mezmaiskaya 2, Les Cottés Z4-1514* and *Spy 94*a). As noted above, only one random allele was considered for the low coverage genomes, in contrast to two for the high-coverage genomes. Positions with more than one missing allele among early Neandertals (out of four alleles) or three missing alleles (out of six) among the late Neandertals were filtered out. We further removed transitions and positions less than 50kb away from the previously

ascertained position. We then computed the proportion of those positions where at least three of the later individuals (i.e. those that lived 40-50kya) carried the derived allele.

## Detecting evidence of Neandertal gene flow into non-Africans

Twelve non-African genomes from the HGDP dataset share a 96 to 139ky-old common ancestor with *Chagyrskaya 8* in the *KNL1* region chr15:40,818,035–41,094,166 in hg19 coordinates (inferred with *Equation 1* as described in Estimating the time of gene flow). Gene flow is necessary to explain this recent common ancestor, but determining the direction of this gene flow is complicated because *Chagyrskaya 8* inherited a modern human-like haplotype in this region. To test whether these 12 non-Africans inherited this copy from Neandertals, we looked for evidence of putative archaic alleles linked to the haplotype in 40 kb windows adjacent to the segment that is modern human-like in *Chagyrskaya 8* (i.e. chr15:40,778,035–40,818,035 and chr15:41,094,166–41,134,166, in hg19 coordinates). We defined these putative archaic variants as those shared with at least one Neandertal genome but absent from Africans. We used vcftools (*Danecek et al., 2011*) to compute the frequency of derived alleles in non-Africans and Africans, revealing that for seven positions in the downstream region (chr15:41,094,166–41,134,166, hg19) the twelve individuals are the only modern humans (out of 929) with a derived allele seen in Neandertal genomes (*Figure 4—figure supplement 1*).

To check whether these is evidence of rare ancestral alleles at positions with missense changes in spindle-associated genes, we used the browser of the *gnomAD* database (v2.1.1, *Karczewski et al., 2020*), which contains 125,748 exome sequences and 15,708 whole-genome sequences from unrelated individuals.

## Acknowledgements

We thank Felipe Mora-Bermúdez, Wieland Huttner, Divyaratan Popli and Hugo Zeberg for helpful discussions and Alba Bossoms Mesa and Leonardo N Iasi for suggestions for the figures. This work was supported by the Max Planck Society, the European Research Council [694707] and the NOMIS Foundation.

## Additional information

### Funding

| Funder | Grant reference number | Author |
| --- | --- | --- |
| Max Planck Society | | Stéphane Peyrégne<br>Janet Kelso<br>Svante Pääbo<br>Benjamin M Peter |
| European Research Council | 694707 | Svante Pääbo |
| NOMIS Foundation | | Svante Pääbo |

The funders had no role in study design, data collection and interpretation, or the decision to submit the work for publication.

### Author contributions

Stéphane Peyrégne, Conceptualization, Formal analysis, Investigation, Methodology, Validation, Visualization, Writing – original draft, Writing – review and editing; Janet Kelso, Benjamin M Peter, Conceptualization, Resources, Supervision, Writing – review and editing; Svante Pääbo, Conceptualization, Funding acquisition, Methodology, Project administration, Resources, Supervision, Writing – review and editing

### Author ORCIDs

Stéphane Peyrégne http://orcid.org/0000-0002-9823-9102
Janet Kelso http://orcid.org/0000-0002-3618-322X
Benjamin M Peter http://orcid.org/0000-0003-2526-8081

Svante Pääbo http://orcid.org/0000-0002-4670-6311

**Decision letter and Author response**
Decision letter https://doi.org/10.7554/eLife.75464.sa1
Author response https://doi.org/10.7554/eLife.75464.sa2

## Additional files

### Supplementary files
• Transparent reporting form

### Data availability
The current manuscript is a computational study, so no data have been generated for this manuscript. Software used are freely and widely available from previous publications and original statistical analyses are described in the main text and the Appendices.

The following previously published datasets were used:

| Author(s) | Year | Dataset title | Dataset URL | Database and Identifier |
|---|---|---|---|---|
| Gene Ontology Consortium | 2021 | The human Gene Ontology Annotation and the basic Gene Ontology terms | http://current.geneontology.org/products/pages/downloads.html | The Gene Ontology resource, goa-human, go-basic |
| Liu X, Li C, Mou C, Dong Y, Tu Y | 2020 | dbNSFP | ftp://dbnsfp@dbnsfp.softgenetics.com/dbNSFP4.2a.zip | dbNSFP, version 4.2 |
| The 1000 Genomes Project Consortium | 2015 | 1000 Genomes Project | http://ftp.1000genomes.ebi.ac.uk/vol1/ftp/phase3/ | The International Genome Sample Resource, Phase 3 |
| Bergstrom A | 2020 | The Human Genome Diversity Panel | ftp://ngs.sanger.ac.uk/production/hgdp | The International Genome Sample Resource, HGDP |

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

# Appendix 1

**Appendix 1—table 1.** Location and predicted effects of the studied amino acid changes in spindle proteins, as reported in dbNSFP version 4.2 (48).

The predictions are for the ancestral variants. We put "damaging" in between quotation marks as the ancestral versions of ATRX and KATNA1 are unlikely to be damaging (as the ancestral amino acid residues are found in the proteins of many species), but that prediction rather supports a function for these amino acid changes.

| protein | position | amino acid change | protein domain (Uniprot) | effect prediction for the ancestral variant (MutPred) | potentially "damaging" ancestral variant according to: |
|---|---|---|---|---|---|
| ATRX | 475 | D ->H | - | - | FATHMM; M-CAP |
| KATNA1 | 343 | A ->T | - | - | FATHMM; PrimateAI |
| KIF18A | 67 | R ->K | kinesin motor (11-355) | Loss of methylation (*P*=0.0087) | - |
| | 159 | H ->R | interaction domain with BUB1 and BUB1B (1-728) | - | - |
| KNL1 | 1,086 | G ->S | 2 × 104 AA approximate repeats (855–1201) | Loss of phosphorylation (*P*=0.0382) | - |
| NEK6 | 291 | D ->H | protein kinase domain (45-310) | - | - |
| RSPH1 | 213 | K ->Q | - | - | - |
| | 43 | P ->S | - | Loss of phosphorylation (*P*=0.0244) Gain of catalytic residue (*P*=0.0179) | - |
| | 162 | E ->G | - | - | - |
| SPAG5 | 410 | D ->H | - | - | - |
| STARD9 | 3,925 | A ->T | - | - | - |

**Appendix 1—table 2.** Deleteriousness and conservation scores at the studied positions with missense changes in spindle genes, as reported in dbNSFP version 4.2 (48).

A high CADD score indicates that the ancestral variant is likely to be deleterious (*Kircher et al., 2014*; *Rentzsch et al., 2019*; *Pollard et al., 2010*) and a high conservation score means that the nucleotide position is highly conserved across species (100 vertebrates for phyloP and phastCons (*Siepel et al., 2005*), and 34 mammals for GERP ++RS, *Davydov et al., 2010*). In contrast to the other scores that correspond to a single position, phastCons is a measure of the conservation in the region around the position. In dbNSFP, the scores range from –6.458163–18.301497 for CADD, from –12.3–6.17 for GERP ++RS, from –20.0–10.003 for phyloP and from 0 to 1 for phastCons.

| | | | Deleteriousness | Conservation | | |
|---|---|---|---|---|---|---|
| gene | position (hg19) rs ID | corresponding amino acid change | CADD score (hg19) | GERP ++RS score | phyloP 100way vertebrate score | phastCons 100way vertebrate score |
| *KATNA1* | 6–149,918,766 rs73781249 | A343T | 2.051033 | 5.48 | 4.834 | 1.000 |

*Appendix 1—table 2 Continued on next page*

*Appendix 1—table 2 Continued*

| gene | position (hg19) rs ID | corresponding amino acid change | Deleteriousness | Conservation | | |
|---|---|---|---|---|---|---|
| | | | CADD score (hg19) | GERP ++RS score | phyloP 100way vertebrate score | phastCons 100way vertebrate score |
| | 17–26,925,570 NA | P43S | –0.425670 | –3.57 | –1.404 | 0.000 |
| | 17–26,919,777 NA | E162G | 0.296317 | 3.66 | 0.280 | 0.001 |
| SPAG5 | 17–26,919,034 NA | D410H | 1.062743 | 5.4 | 2.032 | 1.000 |
| | 15–40,912,860 rs755472529 | H159R | 0.475454 | 3.7 | –0.016 | 0.001 |
| KNL1 | 15–40,915,640 NA | G1086S | 0.801787 | 4.12 | 1.026 | 0.054 |
| KIF18A | 11–28,119,295 rs775297730 | R67K | 1.134589 | 2.62 | 0.525 | 0.845 |
| NEK6 | 9–127,113,155 rs146443565 | D291H | –0.293112 | –1.56 | 3.284 | 1.000 |
| ATRX | X-76,939,325 rs146863015 | D475H | –0.141606 | 3.64 | 0.791 | 0.840 |
| RSPH1 | 21–43,897,491 rs146298259 | K213Q | –0.061053 | 1.81 | 0.672 | 0.079 |
| STARD9 | 15–42,985,549 rs573215252 | A3925T | –0.351117 | –2.51 | 0.047 | 0.000 |

# Appendix 2

**Appendix 2—table 1.** Age estimates of the missense substitutions in spindle genes.
The ages were estimated in the regions where the *Altai Neandertal* and *Denisova 3* genomes fall outside the human variation (intersection of the regions identified with the African-American and deCODE recombination maps). The lower age corresponds to the mean age of the ancestor of multiple present-day African chromosome pairs. The upper age corresponds to the mean age of the common ancestor shared between each present-day African chromosome and either the archaic genome with the least number of differences (excluding *Chagyrskaya 8* for *KNL1*) or a present-day human with an ancestral version of the missense variant(s).

| Gene | chromosome | Region (hg19) | Lower age (kya) | Upper age (kya) |
|------|-----------|---------------|-----------------|-----------------|
| *ATRX* | X | 76,703,773–77,246,471 | NA | NA |
| *KATNA1* | 6 | 149,840,973–149,930,425 | 863 | 1,329 |
| *KIF18A* | 11 | 28,018,167–28,304,293 | 843 | 1,006 |
| *KNL1* | 15 | 40,898,141–40,948,306 | 1,027 | 1,690 |
| *NEK6* | 9 | 127,109,510–127,113,614 | NA | NA |
| *RSPH1* | 21 | 43,897,417–43,897,549 | NA | NA |
| *SPAG5* | 17 | 26,875,942–27,045,524 | 677 | 796 |
| *STARD9* | 15 | 42,941,540–42,989,160 | 947 | 1,401 |

# Appendix 3

**Appendix 3—table 1.** Coverage depth in archaic human genomes at positions with modern human-specific missense substitutions in spindle genes.

The numbers of DNA fragments carrying a particular base are reported in parentheses after the corresponding base. Bases in uppercase were sequenced in the forward orientation, whereas those in lowercase were sequenced in the reverse orientation. Bases that are modern human-like (derived) are highlighted with an asterisk and may represent present-day human DNA contamination or an allele shared with modern humans. The bases that are compatible with a damage-induced substitution (from the ancestral allele) are highlighted in bold (*i.e.*, C-to-T and G-to-A in the forward and reverse orientation, respectively).

| Gene | ATRX | KATNA1 | KIF18A | KNL1 | | NEK6 | RSPH1 | SPAG5 | | | STARD9 |
|---|---|---|---|---|---|---|---|---|---|---|---|
| Chr-Position | X-76,939,325 | 6–149,918,766 | 11–28,119,295 | 15–40,912,860 | 15–40,915,640 | 9–127,113,155 | 21–43,897,491 | 17–26,919,034 | 17–26,919,777 | 17–26,925,578 | 15–42,985,549 |
| Ancestral | C | C | C | A | G | G | T | C | T | G | G |
| Derived | G | T | T | G | A | C | G | G | C | A | A |
| *Altai Neandertal* | C (21)<br>c (31) | C (22)<br>c (15)<br>**T\* (2)** | C (19)<br>c (41)<br>**T\* (1)** | A (24)<br>a (28) | G (33)<br>g (16) | G (23)<br>g (23) | T (18)<br>t (26) | C (17)<br>c (17) | T (18)<br>t (20)<br>a (1) | G (28)<br>g (17) | G (13)<br>g (16)<br>A\* (1) |
| *Chagyrskaya 8* | C (10)<br>c (9)<br>**T (1)**<br>t (1) | C (13)<br>c (5) | C (15)<br>c (19)<br>**T\* (3)** | A (1)<br>G\* (14)<br>g\* (11) | A\* (19)<br>a\* (8) | G (5)<br>g (7)<br>**a (1)** | T (17)<br>t (13) | C (11)<br>c (7)<br>**T (1)** | T (16)<br>t (11) | G (7)<br>g (6)<br>**a\* (1)** | G (8)<br>g (13) |
| *Denisova 3* | C (19)<br>c (17) | C (15)<br>c (13)<br>**T\* (2)** | C (17)<br>c (24) | A (20)<br>a (30) | G (25)<br>g (17)<br>**a\* (1)** | G (15)<br>g (14) | T (19)<br>t (27) | C (20)<br>c (14) | T (16)<br>t (11) | G (12)<br>g (6) | G (17)<br>g (12) |
| *Denisova 11* | C (1)<br>c (2) | NA | C (1) | a (2) | g (1) | G (1)<br>g (1) | T (4)<br>t (2) | c (2)<br>t (1) | NA | NA | G (1) |
| *Goyet Q56-1* | C (1) | NA | C (1)<br>c (5) | G\* (1) | A\* (1)<br>a\* (2) | G (1)<br>g (2) | NA | C (1) | T (1) | G (4)<br>g (2) | G (2) |
| *Hohlenstein-Stadel* | NA | NA | NA | NA | NA | NA | NA | NA | NA | NA | NA |
| *Les Cottés Z4-1514* | C (1)<br>c (1) | c (2)<br>**T\* (1)** | C (5)<br>c (4) | A (2)<br>a (4) | G (3)<br>g (2) | NA | T (2)<br>t (1) | NA | T (1) | G (1) | G (1) |
| *Mezmaiskaya 1* | c (3) | NA | C (1)<br>c (1) | NA | G (2) | NA | t (2) | C (2)<br>c (1)<br>g\* (2) | NA | NA | NA |
| *Mezmaiskaya 2* | C (1) | C (1)<br>c (1)<br>**T\* (1)** | C (1) | g\* (1) | A\* (1) | G (2)<br>g (1) | NA | C (1) | T (1) | G (2) | g (3) |
| *Scladina I-4A* | NA | NA | NA | NA | NA | NA | NA | NA | NA | NA | NA |
| *Spy 94 A* | NA | c (1) | C (1) | g\* (1) | A\* (1)<br>a\* (3) | NA | T (1) | C (1) | T (1) | g (1) | NA |
| *Vindija 33.19* | C (16)<br>c (18)<br>**T (1)** | C (8)<br>c (11) | C (17)<br>c (20)<br>**T\* (1)** | A (13)<br>a (19) | G (20)<br>g (16)<br>**a\* (2)** | G (8)<br>g (8) | T (22)<br>t (15) | C (12)<br>c (14) | T (15)<br>t (7) | G (15)<br>g (13)<br>**a\* (1)** | G (14)<br>g (14) |

**Appendix 3—table 2.** Coverage depth of the *Mezmaiskaya 1* genome at positions with modern human-specific substitutions in *SPAG5*.

Only positions covered by at least one DNA sequence are reported. Bases in uppercase were sequenced in the forward orientation, whereas those in lowercase were sequenced in the reverse orientation. The numbers of DNA fragments carrying a particular base are reported in parentheses after the corresponding base.

| Neandertal | Chr-position (rs ID) | Ancestral | Derived | Allele counts |
|---|---|---|---|---|
| *Mezmaiskaya 1* | 17–26,864,608 (rs188710272) | A | G | A (1) |
| | 17–26,891,162 (NA) | T | G | T (1) |
| | 17–26,892,376 (NA) | A | T | A (2) |
| | 17–26,913,024 (NA) | A | G | a (a) |
| | 17–26,919,034 (NA) | C | G | C (2) c (1) g (2) |
| | 17–26,948,236 (NA) | G | A | g (1) |
| | 17–26,967,723 (rs558276956) | A | G | A (3) |
| | 17–27,005,275 (NA) | G | A | G (1) |
| | 17–27,010,483 (NA) | G | A | g (1) |

# Appendix 4

**Appendix 4—table 1.** Positions defining the closely related haplotype between some modern humans and Neandertals.

At these positions, the *Chagyrskaya 8* genome differs from other high-quality archaic genomes without the modern human-like haplotype but some African genomes from the HGDP dataset carry the same allele as *Chagyrskaya 8*. Note that the modern human-like haplotype in Neandertals is longer and defined by alleles that are shared with all modern humans (*Figure 3*).

| Chromosome | Position (hg19) rs ID | Reference | Alternative (*Chagyrskaya 8*-like) | *Chagyrskaya 8*-like allele frequency in genomes from the HGDP dataset | |
|---|---|---|---|---|---|
| | | | | Africans | Non-Africans |
| 15 | 40,885,107 rs16970851 | A | G | 0.32 | 0.41 |
| | 40,886,017 rs8034043 | C | T | 0.32 | 0.41 |
| | 40,886,020 rs8034048 | C | G | 0.32 | 0.40 |
| | 40,892,601 rs11855923 | G | A | 0.35 | 0.40 |
| | 40,893,573 rs12905162 | C | A | 0.38 | 0.40 |
| | 40,905,450 rs11856438 | C | T | 0.37 | 0.41 |
| | 40,908,904 rs11852670 | A | G | 0.39 | 0.41 |
| | 40,910,707 rs12914743 | T | C | 0.38 | 0.41 |
| | 40,915,045 rs8041534 | T | G | 0.38 | 0.41 |
| | 40,915,894 rs11070285 | T | C | 0.39 | 0.41 |
| | 40,925,214 rs11856802 | T | A | 0.39 | 0.41 |
| | 40,926,654 rs11854986 | C | G | 0.35 | 0.40 |
| | 40,929,814 rs11070286 | T | C | 0.37 | 0.41 |
| | 40,937,647 rs3092979 | A | G | 0.38 | 0.41 |
| | 40,959,413 rs73396515 | G | A | 0.36 | 0.10 |
| | 40,959,624 rs35047458 | G | A | 0.36 | 0.40 |
| | 40,960,432 rs12902568 | G | A | 0.36 | 0.40 |
| | 40,963,160 rs7182530 | A | G | 0.37 | 0.41 |
| | 40,987,528 rs1801320 | G | C | 0.38 | 0.11 |

# Appendix 5

**Appendix 5—table 1.** Origin of the modern human genomes from the HGDP dataset (**Bergström et al., 2020**) with a *KNL1* copy inherited from Neandertals.

| sample | population | region |
| --- | --- | --- |
| HGDP00125 | Hazara | Central South Asia |
| HGDP00547 | Papuan Sepik | Oceania |
| HGDP00639 | Bedouin | Middle East |
| HGDP00696 | Palestinian | Middle East |
| HGDP00714 | Cambodian | East Asia |
| HGDP00774 | Han | East Asia |
| HGDP00822 | Han | East Asia |
| HGDP00954 | Yakut | East Asia |
| HGDP00960 | Yakut | East Asia |
| HGDP00966 | Yakut | East Asia |
| HGDP01023 | Han | East Asia |
| HGDP01181 | Yi | East Asia |

# Appendix 6

**Appendix 6—table 1.** Allele counts at positions with nearly fixed missense variants in the spindle genes of modern humans from the *gnomAD* database (v2.1.1), (***Karczewski et al., 2020***).

Columns 7–8 and 9–10 correspond to the allele counts among the 125,748 whole-exome sequences (WES) and the 15,708 whole-genome sequences (WGS), respectively. Anc = Ancestral

| Gene | Chr-Position (rd ID) | Anc | (nearly) fixed | Alleles | VEP Annot. | # Anc (WES) | Total (WES) | # Anc (WGS) | Total (WGS) |
|---|---|---|---|---|---|---|---|---|---|
| ATRX | X-76,939,325 (rs146863015) | C | G | G-C | missense | 66 | 182,745 | 11 | 22,042 |
| KATNA1 | 6–149,918,766 (rs73781249) | C | T | T-C | missense | 259 | 251,190 | 131 | 31,400 |
| KIF18A | 11–28,119,295 (rs775297730) | C | T | T-C | missense | 26 | 249,508 | 1 | 31,396 |
| | 15–40,912,860 (rs755472529) | A | G | G-A | missense | NA | NA | 1 | 31,368 |
| | | | | G-T | missense | 1 | 227,420 | NA | NA |
| KNL1 | 15–40,915,640 (NA) | G | A | A-G | missense | NA | NA | NA | NA |
| NEK6 | 9–127,113,155 (rs146443565) | G | C | C-G | missense | 164 | 250,140 | 26 | 31,404 |
| | | | | G-T | missense | 236 | 251,414 | 30 | 31,386 |
| RSPH1 | 21–43,897,491 (rs146298259) | T | G | G-A | stop gained | 10 | 251,414 | 1 | 31,386 |
| | 17–26,919,034 (NA) | C | G | G-C | missense | NA | NA | NA | NA |
| | 17–26,919,777 (NA) | T | C | C-A | missense | 3 | 251,430 | NA | NA |
| | 17–26,925,578 (NA) | G | A | A-G | missense | NA | NA | NA | NA |
| SPAG5 | | | | A-T | missense | 1 | 251,066 | NA | NA |
| | 15–42,985,549 (rs573215252) | G | A | A-G | missense | 5 | 139,342 | 3 | 31,284 |
| STARD9 | | | | A-C | missense | 5 | 139,342 | NA | NA |

## Appendix 7

### Derivation of Equation 1

To compute the Time to the Most Recent Common Ancestor (TMRCA) for pairs of modern and archaic humans, we used a published model (***Albers and McVean, 2020***) that we adapted to account for the branch shortening associated with the age of the archaic individual. We simply truncated the posterior distribution of the TMRCA obtained with this model so that the TMRCA cannot be more recent than the age of the archaic individual, and added a correction to account for missing mutations on the archaic branch. Here, we describe how we derived ***Equation 1*** to compute the expected TMRCA with these modifications from the original model.

The expectation of the truncated distribution is:

$E\left(X|X \geq T\right) = \frac{E(X \cap X \geq T)}{P(X \geq T)}$, with $X$ denoting the time to coalescence and $T$ the truncated time,

$= \frac{\int_T^\infty x\, Gamma\left(x|\alpha, \beta\right)\, dx}{\int_T^\infty Gamma\left(x|\alpha, \beta\right)\, dx}$ , with $Gamma\left(x|\alpha, \beta\right)$ denoting the probability density function of the gamma

distribution (with parameters α and β defined as in ***Albers and McVean, 2020***),

$= \frac{\int_T^\infty x^\alpha e^{-\beta x}dx}{\int_T^\infty x^{\alpha-1}e^{-\beta x}dx}$

$= \frac{1}{\beta} \frac{\int_T^\infty (\beta x)^\alpha e^{-\beta x}dx}{\int_T^\infty (\beta x)^{\alpha-1}e^{-\beta x}dx}$

$= \frac{1}{\beta} \frac{\int_{\beta T}^\infty y^\alpha e^{-y}dy}{\int_{\beta T}^\infty y^{\alpha-1}e^{-y}dy}$

$= \frac{1}{\beta} \frac{\Gamma(\alpha+1,\beta T)}{\Gamma(\alpha,\beta T)}$, with $\Gamma\left(\alpha, \beta T\right)$ denoting the upper incomplete gamma function.

By integration by parts, one can show that $\Gamma\left(\alpha + 1, \beta T\right) = \alpha \Gamma\left(\alpha, \beta T\right) + \left(\beta T\right)^\alpha e^{-\beta T}$ to obtain:

$$E\left(X|X \geq T\right) = \frac{1}{\beta}\left(\alpha + \frac{(\beta T)^\alpha e^{-\beta T}}{\Gamma\left(\alpha, \beta T\right)}\right)$$

As this model assumes that the two lineages diverge for the same amount of time (*i.e.*, for a given coalescent time $T$ the total branch length is $2T$), we therefore set $T = \frac{S}{2}$ , with $S$ denoting the age of the ancient specimen, and added $\frac{S}{2}$ as a correction for the branch shortening (*i.e.*, adds $S$ to the total branch length). This leads to ***Equation 1***:

$$E\left(TMRCA\,|TMRCA \geq S\right) = \frac{1}{\beta}\left(\alpha + \frac{\left(\frac{\beta S}{2}\right)^\alpha e^{-\frac{\beta S}{2}}}{\Gamma\left(\alpha, \frac{\beta S}{2}\right)}\right) + \frac{S}{2}$$

## Appendix 8

### Local mutation rate in the region of *KNL1*

For estimating the age of common ancestors shared between modern humans and Neandertals in the region of *KNL1*, we assumed the genome-wide average of $1.45 \times 10^{-8}$ mutations per base pair per generation (*i.e.*, the mutation rate used to estimate the population split times between the ancestors of modern humans and Neandertals *Prüfer et al., 2017*; *Prüfer et al., 2014*; estimated from *Scally and Durbin, 2012*). As the age estimates may be sensitive to the mutation rate used, we also estimated the local mutation rate among family trios from Iceland (*Jónsson et al., 2017*). We counted the number of de novo mutations in the genomes of the probands and divided this number by twice the length of the region and the number of trios in this dataset (1,548) and arrive at a mutation rate estimate of 0.94 [95% Binomial CI: 0.4–1.8] x $10^{-8}$ mutations per base pair per generation in the *KNL1* region where modern humans and *Chagyrskaya 8* share a haplotype. However, there were only 8 de novo mutations among the Icelandic genomes in this region. If we extend that region by 500 kb on both sides to increase the accuracy of the estimate, the mutation rate is 1.1 [95% Binomial CI: 0.9–1.6] x $10^{-8}$ mutations per base pair per generation (45 de novo mutations in this extended region), which is similar to the genome-wide average mutation rate of $1.29 \times 10^{-8}$ estimated in the original study of the trios (*Jónsson et al., 2017*). Therefore, there is no evidence that the mutation rate is different from the genome-wide average in this particular region of the genome.

