## [Editor Report]

Peyrégne et al., studied the genes encoding proteins of the spindle apparatus which exhibit an elevated number of nonsynonymous substitutions in modern humans. In one of these genes (*KNL1*), comparisons of modern and archaic humans identify that some Neanderthals carried the modern human haplotype at the *KNL1* gene, raising the possibility that Neanderthals acquired it from modern humans. Based on the patterns observed in this gene and estimates of the time to the most recent common ancestor, the authors propose an evolutionary scenario that includes an introgression event from modern humans into Neanderthals around 200,000 years ago, and a more recent introgression event from Neanderthals into non-African populations. This study highlights how inspecting individual genomic regions can reveal a complex history of interactions between modern and archaic humans.

---

## [Decision Letter]

**Decision letter after peer review:**

Thank you for submitting your article "The evolutionary history of human spindle genes includes back-and-forth gene flow with Neandertals" for consideration by *eLife*. Your article has been reviewed by 2 peer reviewers (also read by the reviewing editor), and the evaluation has been overseen by a Reviewing Editor and George Perry as the Senior Editor. The following individual involved in review of your submission has agreed to reveal their identity: Christian Huber (Reviewer #2).

Essential revisions:

All reviewers were enthusiastic about the manuscript, and with some revision it will clearly be suitable for publication in *eLife*. There were a number of clarifications that the reviewers felt could strengthen the paper. These included:

1) Comparisons with multiple versions of the variant calling of the Altai Neanderthal genome, and how using different versions could affect the results.

2) The computations of the p-values associated with multiple missense substitutions (see Reviewer 2's comments).

3) What is happening in spindle genes in the human lineage with respect to positive selection and purifying selection? Why did Neanderthals not inherit mutations that were present in the ancestral population of Neanderthals and modern humans (based on their age estimates)? A more explicit discussion of the role of natural selection in explaining the various patterns that are being observed will be helpful.

4) A complex evolutionary history is being proposed for the KNL1 gene. There were parts of the proposed model that require more justification/clarification:

a. Are the estimates of the variants in Neanderthals consistent with estimates from Hubisz et al., 2020?

b. The mutations have estimated ages that are very old, could the authors explain why introgression into Africans from an unrecognized archaic population in Africa or from an ancestral Neanderthal population in Africa is not a more parsimonious explanation than gene flow from humans into Neanderthals?

c. The evidence that the modern-like KNL1 haplotype was introgressed into non-Africans is not that convincing. There is no methods section for the section “Reintroduction of the modern-like KNL1 haplotype into non-Africans.” Who are these 12 individuals who have 7-13 differences? Is the haplotype present outside of Africa very different than the Neanderthal-like haplotype in Africans? Since this haplotype is also present in Africa (as it was called human specific), how do the authors rule out that what is present in those 12 individuals is not just shared variation between non-Africans and Africans? Has this region been previously identified as introgressed in available introgression maps?

d. Why is the ancestral version of KNL1 not found in present-day humans? Wouldn’t this haplotype have been at large frequency in Neanderthals?

Reviewers had a number of other additional suggestions that we felt would improve the manuscript. Please do review and respond to the individual reviewer comments as well.

*Reviewer #1 (Recommendations for the authors):*

There are multiple versions of the variant calling of archaic genomes (e.g., Altai 2013/2014 and Altai 2016 are both being used, but these two versions differ by ~1.5 fold in terms of the numbers of variants). Could the authors comment on how variant calling affects the number of recent human missense substitutions identified in all genes and how it affects their results? I imagine variant calling could significantly impact candidate genes studies like this one. Could the author address and discuss this issue and suggest a best practice for other such candidate genes studies.

Page 3. A p-value for the multiple missense substitutions is provided on page 3, lines 75-76. Based on their Methods, this seems to be computed by random sampling from observed polymorphism. However, this process does not rely on an evolutionary model, and in this test, “substitution event = fixation event” (as they are just random draws) and they behave independently. Yet, on page 4 (line 114), the authors propose using a single fixation event resulting in multiple substitutions to explain the substitutions. In this case, after the first mutation occurred, the second mutation happened in the background with the first mutation, then the two mutations was fixed through one fixation event. Is this much more likely than two independent fixation events? I’d like to see some discussion about it before two independent fixation hypothesis is dismissed. If one fixation is more likely, p-values computed based on the number of “two or more independent draws” does not seem to be valid. This distinction could be quite relevant as the entire survey of the spindle genes is based on spindle genes being statistical outliers.

Could the authors comment more on how they get the segment length and how confident/precise are the segments’ lengths? I only see point estimates for the segment lengths. Is it because a hard threshold is used?

I am a bit surprised that the synonymous substitutions in the spindle genes are not investigated in this study, as they also seem relevant here. Is there any particular technical reasons not working with them? Could the authors provide more information about the current spindle gene polymorphisms within human lineage? Suppose these spindle genes were hotspots in modern human adaptation, we might expect signals of positive selection followed by purifying selection in the human lineage (e.g., in AFR population), realized as πN/πS decrease with time? It seems possible to do this using inferred allele age. This could further justify the study on spindle genes.

The fact that three out of five late Neanderthals carry human-like KNL1 points to one of these possibilities: adaptive introgression, drift, incomplete lineage sorting, and high admixture fraction (with humans). Based on the fact that at least three late Neanderthals also carry 2,773 out of 7,881 possibly introgressed human sites, the author can only dismiss the adaptive introgression hypothesis. How about the others? In particular, does the high frequency contradict the ~3% introgression fraction estimated by Hubisz et al., 2020? If not, how would the authors explain the high observed frequency across all possibly human introgressed sites? Could this suggest contamination or other technical cause?

This study frequently used the length of the segment in testing/estimating gene flow, neutrality, and time. The lengths of the inferred introgressed segments depend on the time of gene flow and the admixture proportion. The observed human to archaic introgression fraction seems much higher than a small percentage and hence may not be negligible when predicting the segment length distribution. Could the authors explain more about whether simplifications/assumptions are involved in their inference? And how these may affect their results.

---

## [Author Response]

Essential revisions:All reviewers were enthusiastic about the manuscript, and with some revision it will clearly be suitable for publication in eLife. There were a number of clarifications that the reviewers felt could strengthen the paper. These included:1) Comparisons with multiple versions of the variant calling of the Altai Neanderthal genome, and how using different versions could affect the results.

We have confirmed that the number of fixed missense variants in the spindle genes does not differ depending on whether the Altai genome is genotyped with GATK or snpAD.

We have used the most up-to-date and accurate genotypes available for all analyses. These are the genotypes generated by snpAD, which models the C-to-T substitutions originating from cytosine deamination correctly and therefore reduces the number of genotyping errors at deaminated sites.

2) The computations of the p-values associated with multiple missense substitutions (see Reviewer 2's comments).

As we describe in the response to Reviewer 2 below, what we test here is if the number of inferred amino acids substitutions in proteins involved in spindle function is unexpected given the overall number of amino acids substitutions in modern humans. This is the case. To us it suggests that spindle function may have changed in modern humans. This is what attracted our interest and motivated the analyses that are presented in the paper.

As the reviewer points out, it is possible, or even likely, that the fixations in *KNL1* and *SPAG5* may go back to single haplotypes that became fixed. We now clarify (lines 75-79; Methods section lines 387-389) that it is the number of amino acid substitutions in this group of functionally related proteins that attract our interest, not the number of genes affected, which is not significantly different from what might be expected by chance (p=0.28). This is illustrated by the fact that only five genes across the genome carry two or more fixed missense substitutions in modern humans, two of them being *SPAG5* and *KNL1.*

3) What is happening in spindle genes in the human lineage with respect to positive selection and purifying selection? Why did Neanderthals not inherit mutations that were present in the ancestral population of Neanderthals and modern humans (based on their age estimates)? A more explicit discussion of the role of natural selection in explaining the various patterns that are being observed will be helpful.

Without direct evidence for selection we feel that any discussion about selection on spindle genes in general would be highly speculative. It is possible and compatible with the data that some new spindle function(s) evolved on the modern human lineage and was positively selected. Perhaps the epistatic interactions among two or more genes was beneficial. Eventually, functional analyses might be able to address these hypotheses (line 354-356).

As for the absence of the derived alleles in archaic humans, even more speculations are possible. If the alleles were at low frequency in the ancestral population, archaic humans may not have inherited them by chance. The ancestral population may have been structured so that the alleles spread mostly in the subpopulation that contributed to modern humans. Alternatively, archaic humans may have inherited the alleles but they could have been lost due to drift, particularly if there were some population bottlenecks or founder effects when archaic humans split from modern humans. Or they may not have been positively selected in the archaic groups due to some other genetic or environmental factor. We do not have any evidence or arguments to prefer any one of these scenarios and would prefer to abstain from what we feel would be excessive speculation.

4) A complex evolutionary history is being proposed for the KNL1 gene. There were parts of the proposed model that require more justification/clarification:a. Are the estimates of the variants in Neanderthals consistent with estimates from Hubisz et al., 2020?

We do not bring any new data regarding the proportion of admixture from modern humans to Neandertals. Therefore, there is no inconsistency with the results from Hubisz et al., 2020.

b. The mutations have estimated ages that are very old, could the authors explain why introgression into Africans from an unrecognized archaic population in Africa or from an ancestral Neanderthal population in Africa is not a more parsimonious explanation than gene flow from humans into Neanderthals?

There are two scenarios raised here:

1: The *KNL1* haplotype would come from an unrecognized archaic group. The age estimate of the common ancestor between the *KNL1* haplotypes of some Neandertals and modern humans is ~200,000 years ago. This makes a scenario where an unknown archaic group would have had to contribute this haplotype to modern humans after 200,000 years ago and it would have spread to become fixed in all modern humans. This group would also have had to be present in Eurasia and contribute this haplotype to Neandertals. This scenario is also made complicated by the fact that the diversity among *KNL1* haplotypes in humans suggest that the age of their MRCA is about 1,027,000 years ago.

2: Could the gene flow have been from early Neandertals to modern humans rather than vice versa. Whereas the *KNL1* substitutions are fixed in modern humans and sit in a region were the MRCA is more than a million years old, the substitutions are present in just some Neandertals and, at least in the Chagyrskaya Neandertal (homozygous derived at those positions), the diversity in the region is small as both copies of the haplotype show few differences. Furthermore, the *KNL1* haplotype with the derived amino acid substitutions is unique across the Neandertal genome in terms of being unusually diverged from the haplotypes without the derived amino acid substitutions (Figure 4A). This is most compatible with gene flow from early modern humans into Neandertals.

c. The evidence that the modern-like KNL1 haplotype was introgressed into non-Africans is not that convincing. There is no methods section for the section “Reintroduction of the modern-like KNL1 haplotype into non-Africans.” Who are these 12 individuals who have 7-13 differences? Is the haplotype present outside of Africa very different than the Neanderthal-like haplotype in Africans? Since this haplotype is also present in Africa (as it was called human specific), how do the authors rule out that what is present in those 12 individuals is not just shared variation between non-Africans and Africans? Has this region been previously identified as introgressed in available introgression maps?

The identifiers of the 12 individuals are provided in Supplement File 2, and we have now added Appendix 5 –Table 1 describing their populations of origin. We have also added a Methods section that describes this analysis (lines 563-579).

There is no evidence for the presence of such haplotype in Africans, as shown in Figure 4C (empty dotted square), and we describe the difference with other haplotypes on lines 289-292. The haplotype in these 12 individuals exhibits 7 to 13 differences to the Chagyrskaya genome. For comparison, other modern human genomes (Africans and non-Africans) exhibit 54 to 179 differences to the Chagyrskaya genome.

As suggested by the reviewer, we checked available introgression maps and the haplotype does not overlap any previously described Neandertal segment in the genomes of present-day people. This is not unexpected given that the haplotype is modern human-like and only seen in the Chagyrskaya Neandertal genome that became available after those maps were generated.

d. Why is the ancestral version of KNL1 not found in present-day humans? Wouldn’t this haplotype have been at large frequency in Neanderthals?

Although this is an interesting observation, the answer to the question “why?” will ultimately depend on functional studies of these changes. At the moment we simply speculate that the ancestral alleles in *KNL1* were perhaps deleterious in modern humans (lines 310-312) and so were lost after contact with Neandertals.

Reviewers had a number of other additional suggestions that we felt would improve the manuscript. Please do review and respond to the individual reviewer comments as well.Reviewer #1 (Recommendations for the authors):There are multiple versions of the variant calling of archaic genomes (e.g., Altai 2013/2014 and Altai 2016 are both being used, but these two versions differ by ~1.5 fold in terms of the numbers of variants). Could the authors comment on how variant calling affects the number of recent human missense substitutions identified in all genes and how it affects their results? I imagine variant calling could significantly impact candidate genes studies like this one. Could the author address and discuss this issue and suggest a best practice for other such candidate genes studies.

We have used the most up-to-date and accurate genotypes available for all analyses. These genotypes were called with snpAD, which is now the standard genotype caller for ancient DNA. Older genotype calls generated with GATK, which does not model ancient DNA damage, *i.e.* C-to-T substitutions originating from cytosine deamination, include more errors as described by Prüfer et al., 2017 and Prüfer et al., 2018.

Page 3. A p-value for the multiple missense substitutions is provided on page 3, lines 75-76. Based on their Methods, this seems to be computed by random sampling from observed polymorphism. However, this process does not rely on an evolutionary model, and in this test, “substitution event = fixation event” (as they are just random draws) and they behave independently. Yet, on page 4 (line 114), the authors propose using a single fixation event resulting in multiple substitutions to explain the substitutions. In this case, after the first mutation occurred, the second mutation happened in the background with the first mutation, then the two mutations was fixed through one fixation event. Is this much more likely than two independent fixation events? I’d like to see some discussion about it before two independent fixation hypothesis is dismissed. If one fixation is more likely, p-values computed based on the number of “two or more independent draws” does not seem to be valid. This distinction could be quite relevant as the entire survey of the spindle genes is based on spindle genes being statistical outliers.

What we test here is if the number of inferred amino acids substitutions in proteins involved in spindle function is unexpected given the overall number of amino acids substitutions seen in modern humans. That this is the case suggests to us that spindle function may have changed in modern humans. This is what attracts our interest and motivates the analyses that are then presented in the paper.

As the reviewer points out, it is possible, or even likely, that the fixations in *KNL1* and *SPAG5* may go back to single haplotypes that became fixed. We now clarify (lines 77-79) that it is number of amino acid substitution in this group of functionally related proteins that attracts our interest, not the number of genes affected, which is not significantly different from what might be expected by chance (p=0.28). This is illustrated by the fact that only five genes across the genome carry two or more fixed missense substitutions in modern humans, two of them being *SPAG5* and *KNL1.*

Could the authors comment more on how they get the segment length and how confident/precise are the segments’ lengths? I only see point estimates for the segment lengths. Is it because a hard threshold is used?

The segments’ length was determined by a hard cutoff on the posterior decoding probability. We assumed that the segment ended when that probability dropped below 0.2 (lines 445-447). As seen in Figure 1, the probability drops rather steeply so that different cutoffs do not substantially change the length of these regions.

I am a bit surprised that the synonymous substitutions in the spindle genes are not investigated in this study, as they also seem relevant here. Is there any particular technical reasons not working with them?

We studied non-synonymous changes because they are more likely to affect the function of the protein.

Could the authors provide more information about the current spindle gene polymorphisms within human lineage? Suppose these spindle genes were hotspots in modern human adaptation, we might expect signals of positive selection followed by purifying selection in the human lineage (e.g., in AFR population), realized as πN/πS decrease with time? It seems possible to do this using inferred allele age. This could further justify the study on spindle genes.

Our focus in this paper is to investigate the evolutionary history of the 11 amino acid changes in the spindle-associated proteins since the split with archaic humans. There is evidence for positive selection around the missense changes in *SPAG5*, *KIF18A* and *ATRX* (lines 106-110 and 129-133). There is also some hint that there could have been negative selection against the ancestral *KNL1* haplotype in modern humans (lines 310-312), but there is no compelling evidence for positive selection for the modern human haplotype in Neandertals.

The fact that three out of five late Neanderthals carry human-like KNL1 points to one of these possibilities: adaptive introgression, drift, incomplete lineage sorting, and high admixture fraction (with humans). Based on the fact that at least three late Neanderthals also carry 2,773 out of 7,881 possibly introgressed human sites, the author can only dismiss the adaptive introgression hypothesis.

We think that there is a misunderstanding here, the 7,881 sites which we used to assess the change in frequency of derived alleles over time in Neandertals do not carry introgressed modern human alleles. They are a random set of sites that happen to be segregating in our sample of Neandertal genomes (some of which have very low coverage, which explains the low number of sites investigated).

How about the others? In particular, does the high frequency contradict the ~3% introgression fraction estimated by Hubisz et al., 2020? If not, how would the authors explain the high observed frequency across all possibly human introgressed sites? Could this suggest contamination or other technical cause?

We believe there is no contradiction with that previous study as mentioned in a response above, since the 2,773 SNPs for which we assess frequency changes in Neandertals are not introgressed from modern humans. We have clarified this in the text (lines 275-276)

This study frequently used the length of the segment in testing/estimating gene flow, neutrality, and time. The lengths of the inferred introgressed segments depend on the time of gene flow and the admixture proportion. The observed human to archaic introgression fraction seems much higher than a small percentage and hence may not be negligible when predicting the segment length distribution. Could the authors explain more about whether simplifications/assumptions are involved in their inference? And how these may affect their results.

We do not estimate the archaic introgression fraction as indicated in the two answers above. Hubisz et al., (2020) estimate 3% and this is low enough that recombination between different introgressed haplotypes is unlikely to be common. If the frequency of the *KNL1* haplotype were high, recombination among copies of this haplotype may bias the estimate based on the haplotype lengths, but note that, in addition to the haplotype lengths, the conclusion of gene flow and its dating rely on the divergence between Neandertal and modern human haplotypes (i.e. lines 257-260).